# From Connectivity to Rewards: Dense Reward Learning with Directed State Graphs

## Abstract

The integration of graphs with Goal-Conditioned Hierarchical Reinforcement Learning (GCHRL) has received increasing attention, as graphs naturally encode task hierarchies for effective subgoal sampling. However, existing methods often overlook intrinsic connectivity information, failing to fully leverage the underlying topology for efficient learning. Most graph-based GCHRL methods use the graph as a stochastic sampling tool rather than as an environmental model that encodes connectivity and state-accessibility information. This limitation is particularly acute in quasimetric environments, where the inherent asymmetry of state transitions poses a fundamental challenge to stable policy learning and robust path planning. In this paper, we address these problems by introducing a state connectivity model designed to predict pairwise state connectivity strength in asymmetric environments. We transform these connectivity strengths into scalar auxiliary dense rewards, providing continuous guidance across multiple hierarchical levels. We demonstrate that our proposed framework, Graph-Guided Quasimetric Dense Reward (G2QDR), can theoretically be integrated into any existing GCHRL architecture, and the state connectivity model is efficiently implemented via a neural network trained on a directed state graph generated during exploration. Empirical results across a wide range of sparse reward environments indicate that, in general, G2QDR can enhance the performance of baseline GCHRL approaches with acceptable computational overhead.

## 1 Introduction

Sparse rewards remain a fundamental challenge in reinforcement learning (RL), where agents must learn from infrequent and delayed feedback signals. In such settings, traditional, non-hierarchical approaches (Schulman et al., 2017; Fujimoto et al., 2018; Haarnoja et al., 2018) often struggle with inefficient exploration and poor credit assignment. Goal-Conditioned Hierarchical Reinforcement Learning (GCHRL) addresses this issue by decomposing a long-horizon task into a sequence of intermediate subgoals: a high-level policy proposes subgoals, while a low-level policy learns to achieve them. This hierarchical decomposition significantly improves learning efficiency by transforming a difficult global objective into a set of more tractable local problems.

Despite these advantages, early GCHRL methods (Nachum et al., 2018a;b) often suffer from sample inefficiency, as the high-level agent must select subgoals from the entire state space—a task that can be even more challenging than selecting actions in non-hierarchical approaches. Prior work has attempted to address this issue by constraining subgoal selection to local regions (Zhang et al., 2022), imposing smoothness constraints on sampled subgoals (Li et al., 2022), or employing stochastic sampling strategies (Wang et al., 2024; 2025). While these approaches reduce the burden on the high-level agent, they do not organize visited states or subgoals into a structured representation, thereby losing important connectivity information.

In contrast, graph-based GCHRL methods (Lee et al., 2022; Gieselmann & Pokorny, 2021) model relationships and connectivity between states explicitly through a state graph, which is inherently well-suited for representing the environment and task structure. Previous graph-based work has explored directions such as decision-making via graph search or traversal (Wan et al., 2021; Shang et al., 2019; Eysenbach et al., 2019),

as well as the use of graphs as world models (Zhang et al., 2021; Huang et al., 2019). However, many of these approaches rely on pre-crafted graphs, which limit their generalizability. Moreover, most existing methods (Zhu et al., 2022; Hong et al., 2022) operate directly on the constructed graph, making them less effective when the agent encounters states that are not represented in the graph.

Graph learning, alternatively, enables the agent to leverage relational structure even when the current state is not explicitly represented in the graph, by generalizing through learned embeddings and connectivity patterns. Recent studies (Zhang et al., 2025) have shown that incorporating graph representations into the learning process can significantly enhance the performance of underlying reinforcement learning algorithms, improving both sample efficiency and generalization. Moreover, graph learning (Klissarov & Precup, 2020; Klissarov & Machado, 2023) facilitates more informed exploration and planning by capturing the topology of the state space, allowing agents to reason about long-range dependencies and transitions beyond their immediate experience.

Another common challenge in graph-based methods is that distances between states are often quasimetric (Wang & Isola, 2022; Wang et al., 2023), meaning that transition costs are asymmetric: moving from B to A can be significantly more difficult than moving from A to B. Undirected state graphs (Zhang et al., 2025) fail to capture this asymmetry, motivating the use of directed graphs instead.

In this paper, we propose a graph-based online GCHRL framework that addresses these challenges. We construct a directed state graph online during exploration, incrementally incorporating visited states while pruning outdated nodes and connections. We then train a model on this graph to predict state connectivity, which serves as a proxy for transition distance. This connectivity measure is subsequently used to generate dense reward signals, enabling more efficient learning and improved understanding of the environment.

The main contributions of this paper are as follows:

- We introduce a method for the online construction of a directed state graph as a graph-based environmental model (Ha & Schmidhuber, 2018; Zhang et al., 2021). The graph is built directly during exploration, eliminating the need for expert data or manually designed structures.

- We propose a novel architecture that incorporates a state connectivity model to predict empirical connectivity between states based on observed arrivals. This model allows the evaluation of newly visited states that are not yet represented in the graph.

- We leverage auxiliary rewards (Simsek & Barto, 2006; Nehmzow et al., 2013), derived from state connectivity, to improve learning efficiency for the high-level agent and to provide well-calibrated learning signals for the low-level agent.

- Our architecture is theoretically compatible with any GCHRL algorithm in both symmetric and asymmetric environments. In our experiments, we evaluate it with four representative backbone methods (HIRO, HRAC, HESS, and HLPS), and observe performance improvements across a broad range of tasks.

We evaluated our approach across a range of MuJoCo environments (Todorov et al., 2012) to assess the significance of our experimental results. The results show that our method improves the performance of the underlying GCHRL framework in terms of success rate.

## 2 Preliminaries

### 2.1 Markov Decision Processes

As the most widely used framework for modeling reinforcement learning problems, the Markov Decision Process (MDP) (Puterman, 2014) is defined as a tuple $\langle \mathcal{S}, \mathcal{A}, P, R, \gamma \rangle$. At each time step $t$, the agent observes the current state $s_t \in \mathcal{S}$ provided by the environment and selects an action $a_t \in \mathcal{A}$ according to its policy $\pi(a_t \mid s_t)$, which specifies the probability of choosing action $a_t$ given state $s_t$.

Once the action is executed, the environment transitions the agent to a new state $s_{t+1}$ according to the transition probability function $P(s_{t+1} \mid s_t, a_t)$, which is typically unknown in model-free settings. The agent then receives a reward $r_t$ determined by the reward function $R(s_t, a_t)$, which evaluates the action taken in the current state and is also not directly accessible to the agent.

The objective of the agent is to learn an optimal policy $\pi$ that maximizes the expected discounted cumulative reward $\mathbb{E}_\pi\left[\sum_{t=0}^{T} \gamma^t r_t\right]$, where $\gamma \in [0, 1)$ is a predefined discount factor that prioritizes immediate rewards over those in the distant future, ensuring the total return remains finite.

## 2.2 Goal-conditioned Hierarchical RL (GCHRL)

Goal-Conditioned Reinforcement Learning (GCRL) (Liu et al., 2023) extends standard reinforcement learning by training agents to achieve specific goals, typically defined as target states. In this setting, the agent receives an additional goal input $g_t$ alongside the state $s_t$ and learns a goal-conditioned policy $\pi(a_t \mid s_t, g_t)$ that aims to reach the desired goal. By explicitly incorporating goals into the policy input, the agent's behavior is guided toward achieving specified outcomes. The reward function is often goal-dependent, providing positive feedback when the agent successfully reaches the target state.

To address the challenges posed by large and complex environments, Goal-Conditioned Hierarchical Reinforcement Learning (GCHRL) (Nachum et al., 2018b; Zhang et al., 2022; Wang et al., 2024) decomposes the task into a hierarchy of simpler and more manageable sub-tasks. Typically, this framework consists of two levels of control. For every $p$ steps, the high-level agent selects a subgoal $g_t$, representing an intermediate target state, which is then passed to a low-level agent for execution. The subgoal is sampled from a high-level policy $\pi_h(g_t \mid \phi(s_t))$, where $\phi : \mathcal{S} \to \mathbb{R}^d$ denotes a state representation function that maps the raw state to a compact feature space.

Given the subgoal $g_t$ and the state representation $\phi(s_t)$, the low-level policy selects an action $a_t$ according to the policy $\pi_l(a_t \mid \phi(s_t), g_t)$. The low-level policy is trained using an intrinsic reward signal defined as $r_{\text{int}}(s_t, g_t, a_t, s_{t+1}) = -\|\phi(s_{t+1}) - g_t\|_2$, which encourages the agent to minimize the distance between the achieved state representation and the subgoal.

Both the high-level and low-level agents can be implemented using policy-based reinforcement learning methods, including those developed in prior work on policy gradients, such as (Fujimoto et al., 2018; Haarnoja et al., 2018; Schulman et al., 2017).

## 2.3 Graph Abstraction of MDP

A graph is a general and flexible data structure for modeling complex relationships among objects in many real-world problems. A graph is defined as $\mathcal{G} = (\mathcal{V}, \mathcal{E})$, where $|\mathcal{V}| = N$ denotes the set of nodes and $\mathcal{E} = \{e_{ij}\}$ denotes the set of edges (without self-loops). The adjacency matrix of $\mathcal{G}$ is given by $\boldsymbol{A} = (\boldsymbol{A}_{i,j}) \in \mathbb{R}^{N \times N}$, where $\boldsymbol{A}_{i,j} = 1$ if there exists an edge between nodes $i$ and $j$, and $\boldsymbol{A}_{i,j} = 0$ otherwise.

This formulation can be naturally extended to a weighted adjacency matrix, in which $\boldsymbol{A}_{i,j}$ represents the weight associated with edge $e_{ij}$, capturing the strength or significance of the relationship.

In the context of Markov Decision Processes (MDPs), nodes can be interpreted as states, while edge weights encode transition probabilities or reachability statistics between states. Under this perspective, the graph serves as an abstract representation of the environment dynamics, capturing how states are interconnected through agent–environment interactions (Lee et al., 2022).

More generally, such a graph can be viewed as a compressed or learned structural model of the MDP (Zhang et al., 2025), where the connectivity reflects feasible transitions and the edge weights quantify their likelihood or frequency. This abstraction is particularly useful in large or continuous state spaces, where explicitly modeling the full transition function is intractable. By operating on the graph structure, one can exploit relational information between states to facilitate planning, exploration, and representation learning.

# 3 Methods

Previous work (Zhang et al., 2025) focuses on deriving dense rewards from undirected graphs, but this approach encounters difficulties in environments with directed connectivity, where state symmetry does not hold. In this section, we present our framework, Graph-Guided Quasimetric Dense Reward (G2QDR)[1], which explicitly models the state connectivity with a directed graph. The overall structure of our framework within the traditional GCHRL pipeline is illustrated in Figure 1.

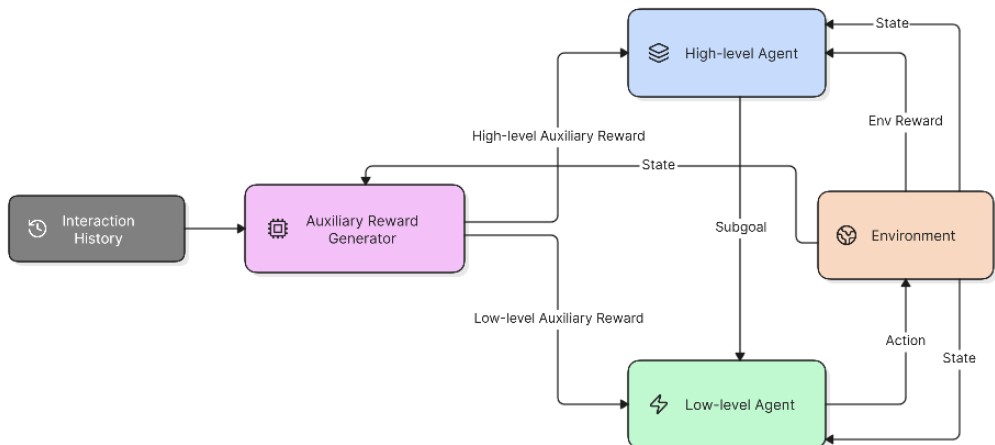

Figure 1: **Overview of the G2QDR framework.** G2QDR serves as an auxiliary reward generator that takes the environmental state and interaction history as inputs, producing auxiliary rewards at multiple levels to facilitate the learning of both the high-level and low-level policies.

By integrating auxiliary rewards derived from state transitions with the original sparse rewards for both high- and low-level policies, the policy learning process more rapidly adapts to the underlying state connectivity structure. This improves learning efficiency by encouraging more plausible subgoal proposals and more effective use of implicit spatial information.

## 3.1 State graph

To record the visited states and their relationships, we maintain a directed state graph $\mathcal{G} = (\mathcal{V}, \mathcal{E})$ with a fixed number $N$ of nodes[2]. This graph is built incrementally during the exploration phase of training, without relying on expert data or a handcrafted process.

Each node is associated with a state. For a node corresponding to state $s_t$, we store the corresponding state representation vector $\phi(s_t) \in \mathbb{R}^d$ as the node feature and use $s_t$ as the node label for simplicity. The edges in the graph then represent the connectivity between states.

We model the state graph as directed to capture the potentially asymmetric nature of transitions in directed connectivity environments; directed edges naturally encode these directional discrepancies.

### 3.1.1 Incremental graph construction

The graph is initialized with $N$ empty nodes and no edges. The corresponding weighted adjacency matrix $\boldsymbol{A}$ is set to an $N \times N$ zero matrix. We explore the environment using the backbone agent with randomly initialized high- and low-level policies $\pi_h$ and $\pi_l$. Whenever the agent encounters a previously unseen state representation—i.e., one that is sufficiently distant from all state representations currently stored in the

---

[1]The term 'quasimetric' in the method name does not imply that our approach explicitly or implicitly employs a quasimetric model. Rather, it emphasizes the underlying motivation of the method: enabling the agent to distinguish between opposite directions of transition between states.

[2]The number of training examples for the state connectivity model grows quadratically with $N$ because the adjacency weight matrix has $N^2$ elements. The choice of $N$ depends on the machine's capabilities.

graph, as defined in Equation (1), the state is added to the graph as a node, with the representation $\phi(s_t)$ as its feature. We define a window of size $W$, such that the $W$ most recent preceding states are considered connected to the current state. Accordingly, we create edges between the current node and the nodes corresponding to these preceding states, as shown in equation (2).

$$\forall_{s_v \in \mathcal{V}}, \|\phi(s_t) - \phi(s_v)\|_2 > \epsilon_d, \tag{1}$$

$$\boldsymbol{A}_{s_{t-w}, s_t} = w^{-p}, \quad w = 1, \ldots, W, \tag{2}$$

where $\epsilon_d$ is a hyperparameter that specifies the distance threshold between state representations, and $p > 0$ is a hyperparameter that controls the polynomial decay's exponent. Naturally, we initialize the edge weight using the decay $w^{-p}$, as states that are farther apart in the trajectory (i.e., larger $w$) are considered more weakly connected.

When the agent encounters a state $s_t$ with representation $\phi(s_t)$ that is similar to some node features already stored in the graph, it finds the state whose representation is the closest to the current state feature:

$$s_v = \underset{s_u : \|\phi(s_t) - \phi(s_u)\|_2 \leq \epsilon_d}{\arg\min} \|\phi(s_t) - \phi(s_u)\|_2. \tag{3}$$

Then, we treat the state $s_t$ as $s_v$, relabel the node $s_v$ as $s_t$, and relabel the corresponding edges accordingly. The weight for the edge $(s_{t-w}, s_t)$ is then updated as follows:

$$\boldsymbol{A}_{s_{t-w}, s_t} := \boldsymbol{A}_{s_{t-w}, s_t} + w^{-p}, \quad w = 1, \ldots, W. \tag{4}$$

Note that larger weights correspond to more frequent transitions between the underlying states. Conversely, for states that are farther apart in the trajectory (i.e., with larger $w$), the increment $w^{-p}$ is smaller, reflecting the weaker connection between them.

We use the Euclidean norm to measure the distance between representation vectors. However, since some components may encode more spatial information than others, a weighted Euclidean norm can alternatively be employed to define distances between state representations, allowing the metric to better reflect the structure of the environment.

### 3.1.2 Updating in a full graph

The graph has a fixed number of nodes. Suppose the graph is full, when a new state $s_t$ is encountered, if $s_v$ from equation (3) exists, as before we relabel the node as $s_v$ and perform edge update as shown in equation (4); Otherwise, we replace the oldest state node in the graph with the current state node, delete all edges previously linked to that node, and initialize the edges using equation (2). Alternatively, one could replace the node that is most weakly connected to the rest of the graph—that is, the node with the lowest sum of edge weights.

### 3.2 State connectivity model

To assign appropriate connectivity strengths to all possible state pairs, including those not yet observed, we use node features and edges to train a state connectivity model, which predicts connectivity strengths between unseen states whose representations are not stored in the graph.

The model $\mathcal{C}_\theta$ maps every paired state representation $\psi(s_u, s_v)$, which is a column vector, to a connectivity score $c$, where $\theta$ denotes its parameters. We instantiate $\mathcal{C}_\theta$ as a multi-layer feed-forward network (FFN):

$$c = \mathcal{C}_\theta(\psi(s_u, s_v)) = \text{FFN}(\psi(s_u, s_v)). \tag{5}$$

The state connectivity model and the policies are updated alternately during policy training.

**Order-sensitive paired state representation** Since transition distances between states are often asymmetric, the paired-state representation should be order-dependent to enable learning asymmetric state connectivities (i.e. $\mathcal{C}_\theta(\psi(s_u, s_v)) \neq \mathcal{C}_\theta(\psi(s_v, s_u))$). Here, we propose some candidates for order-sensitive paired state representation:

- **Vector Concatenation** The input $\psi(s_u, s_v) = \begin{bmatrix} \phi(s_u) \\ \phi(s_v) \end{bmatrix} \in \mathbb{R}^{2d}$ is just a simple concatenation of the two state representation vectors. In this case, the intermediate layers of $\mathcal{C}_\theta$ learn higher-level features of the paired input $\psi(s_u, s_v)$.

- **Gated Fusion** The input $\psi(s_u, s_v) \in \mathbb{R}^d$ is computed via gated fusion parameterized by learnable weights $\boldsymbol{W} \in \mathbb{R}^{d \times 2d}$ and biases $\boldsymbol{b} \in \mathbb{R}^d$:

$$g(s_u, s_v) = \sigma(\boldsymbol{W} \begin{bmatrix} \phi(s_u) \\ \phi(s_v) \end{bmatrix} + \boldsymbol{b}) \in \mathbb{R}^d, \ \psi(s_u, s_v) = g(s_u, s_v) \odot \phi(s_u) + (1 - g(s_u, s_v)) \odot \phi(s_v), \quad (6)$$

  where $\sigma$ is a gating function. Gated fusion learns the contribution of each input at the feature level. Instead of enforcing a fixed combination, it enables data-dependent weighting between the two inputs. Consequently, the representations $\psi(s_u, s_v)$ and $\psi(s_v, s_u)$ are distinctly different, rather than being related by a simple permutation.

**Training loss** Naturally we can use $\boldsymbol{A}_{s_u, s_v}$ as a measure of connectivity between nodes $s_u$ and $s_v$. However, to constrain the scale of the network outputs during training, we instead normalize this quantity as $\boldsymbol{A}_{s_u, s_v} / \max_{s_i, s_j} \boldsymbol{A}_{s_i, s_j}$.

Accordingly, the loss function is defined as:

$$\mathcal{L} = \sum_{\phi(s_u), \phi(s_v) \in \mathcal{V}} \left[ \mathcal{C}_\theta(\psi(s_u, s_v)) - \boldsymbol{A}_{s_u, s_v} / \max_{s_i, s_j} \boldsymbol{A}_{s_i, s_j} \right]^2. \quad (7)$$

This loss function encourages the model to preserve the local neighbourhood structure of the graph in its predictions. In doing so, the model learns the empirical connectivity between states as induced by the graph.

Note that in each training phase of the model (except the first), the parameters are initialized using the values learned in the previous phase, which serves as a good initial point.

### 3.3 Induced dense reward

Our proposed method introduces induced dense reward signals across all levels of agents in traditional goal-conditioned settings, enabling better utilization of experienced environmental dynamics and promoting more efficient learning.

**High-level constraint reward** The high-level policy $\pi_h(g_t | \phi(s_t))$ proposes a subgoal every $P$ steps and is trained using the external environmental reward $r_{\text{ext}}$. It can be implemented using any policy-based reinforcement learning algorithm that operates on transition tuples $(s_t, g_t, r_t, s_{t+1})$

To promote more efficient exploration, we encourage the policy to generate subgoals that are easier to reach from the current state $s_t$. Specifically, we augment the high-level reward with an intrinsic term that rewards strong connections between the current state $s_t$ and generated subgoal $g_t$. The resulting reward is defined as follows:

$$r_h(s_t, g_t, s_{t+1}) = r_{\text{ext}} + r_{\text{int}} = r_{\text{ext}} + \alpha_h \cdot \mathcal{C}_\theta(\psi(s_t, g_t)), \quad (8)$$

where $\alpha_h \geq 0$ is a hyperparameter that controls the significance of the intrinsic term in the high-level reward.

**Low-level correction reward**    The low-level policy $\pi_l(a_t|\phi(s_t), g_t)$ can be implemented using any policy-based reinforcement learning algorithm that operates on transition tuples $(s_t, g_t, a_t, r_t, s_{t+1})$. In the traditional low-level policy learning process (Nachum et al., 2018b), the policy learns exclusively from reward signals derived from the Euclidean distance between the subsequent state representation and the subgoal. However, this approach is limited as it ignores environmental connectivity and fails to reflect the (true) transition distance between states accurately. We address this limitation by introducing a connectivity-based term as follows:

$$r_l(s_t, g_t, a_t, s_{t+1}) = -\|\phi(s_{t+1}) - g_t\|_2 + \alpha_l \cdot \mathcal{C}_\theta(\psi(s_{t+1}, g_t)), \tag{9}$$

where $\alpha_l \geq 0$ is a hyperparameter controlling the significance of the reward term in the low-level reward.

**Asymmetric penalty**    In certain environments, asymmetric transitions can prevent an agent from reaching its goal (e.g., one-way dead ends, pits, or cliffs). In such cases, a state $s_v$ may be easily reachable from $s_u$, while the reverse transition from $s_v$ to $s_u$ is significantly more difficult or unlikely. To capture this asymmetry, we compare the learned directed connectivity scores $\mathcal{C}_\theta(\psi(s_u, s_v))$ and $\mathcal{C}_\theta(\psi(s_v, s_u))$, which serve as proxies for transition ease in each direction.

We define penalty terms for both the high- and low-level policies as:

$$r_{hp} = -\alpha_{hp} \cdot \max(\mathcal{C}_\theta(\psi(s_t, g_t)) - \mathcal{C}_\theta(\psi(g_t, s_t)), 0), \quad r_{lp} = -\alpha_{lp} \cdot \max(\mathcal{C}_\theta(\psi(s_t, s_{t+1})) - \mathcal{C}_\theta(\psi(s_{t+1}, s_t)), 0). \tag{10}$$

where $\alpha_{hp} \geq 0$ and $\alpha_{lp} \geq 0$ are hyperparameters, which control the strength of the penalty. The asymmetric gap $\mathcal{C}_\theta(\psi(s_u, s_v)) - \mathcal{C}_\theta(\psi(s_v, s_u))$ quantifies directional irreversibility. A large positive gap indicates that the transition from $s_u$ to $s_v$ is substantially easier than the reverse, suggesting a potentially hazardous or trapping structure in the environment.

The use of the $\max(\cdot, 0)$ operator ensures that only risky asymmetry is penalized. Without this clipping, the term could encourage the agent to favour transitions that are difficult to reach but easy to leave, which is an undesirable and unintended behavior. This penalty encourages both high- and low-level policies to remain within regions of the state space that exhibit more balanced bidirectional connectivity, corresponding to more reversible and safer transitions.

### 3.4    Evolution of dense signal strength

Due to the varying quality of the dense signal across different training stages, we introduce a stage-dependent controller $\lambda$ to modulate the induced reward terms. Specifically, the hyperparameters $\alpha_h$, $\alpha_l$, $\alpha_{hp}$, and $\alpha_{lp}$ are all scaled by $\lambda$. Let $n_t$ denote the current episode, with the activation phase spanning episodes $n_1$ to $n_2$ and the annealing phase spanning episodes $n_3$ to $n_4$. The controller is defined as:

$$\lambda = \begin{cases} 0 & \text{warm-up stage} \\ \frac{n_t - n_1}{n_2 - n_1} & \text{activation stage} \\ 1 & \text{full reward stage} \\ \frac{n_4 - n_t}{n_4 - n_3} & \text{annealing stage} \\ 0 & \text{final stage} \end{cases} \tag{11}$$

The construction of the graph and the training of the state connectivity model follow a bootstrapping process: in early episodes, the graph covers only a limited region of the environment, its edges are poorly calibrated, and the model is correspondingly biased. Therefore, during the warm-up stage, the graph is used to update the model, but the model is not used for policy learning (i.e., $\lambda = 0$), reflecting the limited reliability of the graph and the model.

In the activation stage, we linearly increase $\lambda$ from 0 to 1, gradually introducing the dense signal induced by our model. This additional signal guides more efficient exploration and policy learning. $\lambda$ is fixed to 1 afterwards, in the full reward stage.

In the annealing stage, since the dense reward is not potential-based (Ng et al., 1999), it not only alters the optimal policy but also changes the underlying optimization problem. To realign with the original task, we gradually phase out the dense signal by decreasing $\lambda$ from 1 to 0. In the final stage, $\lambda$ is fixed to 0.

### 3.5   Algorithm: GCHRL + G2QDR

A detailed description of how the proposed G2QDR strategy can be integrated into online GCHRL algorithms is provided in Algorithm 1.

---

**Algorithm 1:** GCHRL+G2QDR

---

**Input:** High-level policy $\pi_h(g \mid \phi(s))$, low-level policy $\pi_l(a \mid \phi(s), g)$, replay buffer $\mathcal{B}$, state connectivity model $\mathcal{C}_\theta$, high-level action frequency $K$, window size $W$, significance thresholds $\alpha_h$, $\alpha_l$, $\alpha_{hp}$, $\alpha_{lp}$, adaptive signal strength $\lambda$, number of episodes $N$, maximum number of steps per episode $T$.

$n \leftarrow 0$;
**while** $n < N$ **do**
    $t \leftarrow 0$;
    **while** $t < T$ **do**
        **if** $t \bmod K = 0$ **then**
            Sample subgoal $g_t \sim \pi_h(\cdot \mid \phi(s_t))$;
        **else**
            Retain the current subgoal $g_t$;
        Sample atomic action $a_t \sim \pi_l(\cdot \mid \phi(s_t), g_t)$;
        Execute $a_t$ and observe reward $r_t$ and next state $s_{t+1}$;
        **if** $\lambda \neq 0$ **then**
            Compute $r_h(s_t, g_t, s_{t+1})$, $r_l(s_t, g_t, a_t, s_{t+1})$, and penalty terms $r_{hp}$ and $r_{lp}$ using Eqs. (8), (9), and (10);
            Store transition $(s_t, g_t, a_t, \lambda, [r_{\text{ext}}, -\|\phi(s_{t+1}) - g_t\|_2, \mathcal{C}_\theta(\psi(s_t, g_t)), \mathcal{C}_\theta(\psi(s_{t+1}, g_t)), r_{hp}, r_{lp}], s_{t+1})$ in replay buffer $\mathcal{B}$;
        **else**
            Store transition $(s_t, g_t, a_t, 0, [r_{\text{ext}}, -\|\phi(s_{t+1}) - g_t\|_2, 0, 0, 0, 0], s_{t+1})$ in replay buffer $\mathcal{B}$;
        Update graph node representations and edge weights using the collected transition and Eqs. (1)–(4);
        Update the state connectivity model $\mathcal{C}_\theta$ using the current graph structure;
        $t \leftarrow t + 1$;
    Update the low-level policy $\pi_l(a \mid \phi(s), g)$ using the selected GCHRL learning algorithm and samples from $\mathcal{B}$;
    Update the high-level policy $\pi_h(g \mid \phi(s))$ using the selected GCHRL learning algorithm and samples from $\mathcal{B}$;
    Update the adaptive signal strength $\lambda$ according to the current episode index $n$;
    $n \leftarrow n + 1$;

---

## 4   Experiments

As is standard in the GCHRL community, we evaluate our method across a range of long-horizon continuous control tasks simulated in the MuJoCo (Todorov et al., 2012) environment. In this section, we employ gated fusion to construct an order-sensitive representation of state pairs.

### 4.1   Environment settings

We adopt the **AntMaze**, **AntGather**, **AntPush**, **AntFall**, and **Pusher** environments from MuJoCo, all configured with sparse reward settings. The first four involve complex navigation and manipulation tasks performed by a simulated multi-limbed robot, whereas **Pusher** focuses on object manipulation using a multi-jointed robotic arm. These environments exhibit varying degrees of asymmetry in their state transitions.

**AntMaze** is largely symmetric, as the agent can relatively easily return to previously visited states. In contrast, **AntGather**, **AntPush**, **AntFall**, and **Pusher** exhibit different types of asymmetry: reversing object displacement in AntGather, AntPush and Pusher is considerably more difficult, and ascending cliffs in AntFall is substantially harder than descending them.

## 4.2 Comparative analysis

We incorporated G2QDR in the following existing GCHRL methods:

- **HIRO** (Nachum et al., 2018b): One of the earliest approaches to demonstrate effective integration of goal-conditioned information in online hierarchical reinforcement learning.

- **HRAC** (Zhang et al., 2022): Extends HIRO by introducing an adjacency network that generates subgoals which are more readily reachable from the current state, thereby improving overall performance.

- **HESS** (Li et al., 2022): This method adds a regularization term over consecutive subgoal representations during each update to promote stability across episodes.

- **HLPS** (Wang et al., 2024): Uses a Gaussian process over subgoal representations to enable smoother and more consistent updates.

In addition to comparing these four G2QDR-augmented methods with their original counterparts, we also evaluate them against the following undirected-graph baseline to highlight the advantages of directed graph learning in asymmetric environments:

- **G4RL** (Zhang et al., 2025): A plug-in state connectivity estimator based on undirected graphs. It employs a graph encoder–decoder to learn the subgoal space and estimate transition plausibility from learned subgoal representations.

We report average success rates on **AntMaze**, **AntPush**, **AntFall**, and **Pusher**, along with the average number of objects collected in **AntGather** (Table 1). All results are averaged over 10 independent runs and reported as the mean ± standard deviation.

The agents are trained for a total of 20000 episodes in each environment. Every 1,000 training episodes, the current policy is evaluated over 100 independent evaluation trials without further learning. The performance metric is the success rate, defined as the percentage of trials in which the agent completes the task. The success rate is recorded after each evaluation to monitor the learning progress.

From Table 1, we observe that incorporating G2QDR into the base GCHRL methods generally improves performance across the evaluated environments. In many cases, the G2QDR-augmented variants achieve higher final success rates than their corresponding baselines while also exhibiting lower performance variance. These improvements are particularly pronounced in tasks that heuristically exhibit a higher degree of asymmetry. Although the gains are not consistent across all environments, the overall trend indicates that G2QDR enhances both the effectiveness and the stability of the underlying methods.

In asymmetric environments, G2QDR typically provides larger improvements over both the original baselines and the undirected G4RL variant. These results indicate that explicitly modeling directional relationships can be particularly beneficial in environments with asymmetric dynamics.

## 4.3 Ablation study

### 4.3.1 Impact of auxiliary reward components

To evaluate the effectiveness of different auxiliary reward components within G2QDR, we consider the following variants:

Table 1: **Main results on MuJoCo environments across different methods.** For each method, we report the performance of the original baseline together with its undirected variant (+G4RL) and directed variant (+G2QDR). The variant **(+G2QDR w/o PT)** applies G2QDR with combined constraints and corrections but without penalty terms, while **(+G2QDR w/ PT)** additionally incorporates penalty terms. All results are reported at the final training episode. Within each method group, the best-performing variant is underlined.

| | AntMaze U-shape | AntMaze W-shape | AntGather | AntPush | AntFall | Pusher |
|---|---|---|---|---|---|---|
| HIRO | 0.72 ± 0.09 | 0.53 ± 0.14 | 1.47 ± 0.18 | 0.05 ± 0.01 | 0.23 ± 0.04 | 0.17 ± 0.03 |
| +G4RL | 0.82 ± 0.07 | 0.61 ± 0.16 | 1.78 ± 0.10 | 0.14 ± 0.02 | 0.28 ± 0.04 | 0.21 ± 0.02 |
| +G2QDR w/o PT | 0.87 ± 0.08 | 0.66 ± 0.09 | 1.90 ± 0.11 | 0.12 ± 0.01 | 0.35 ± 0.03 | 0.26 ± 0.03 |
| +G2QDR w/ PT | 0.80 ± 0.04 | 0.59 ± 0.07 | 1.96 ± 0.13 | 0.19 ± 0.04 | 0.31 ± 0.03 | 0.30 ± 0.03 |
| HRAC | 0.74 ± 0.05 | 0.59 ± 0.27 | 2.03 ± 0.22 | 0.10 ± 0.01 | 0.21 ± 0.06 | 0.24 ± 0.04 |
| +G4RL | 0.90 ± 0.03 | 0.68 ± 0.19 | 2.41 ± 0.26 | 0.12 ± 0.02 | 0.34 ± 0.03 | 0.22 ± 0.02 |
| +G2QDR w/o PT | 0.84 ± 0.03 | 0.70 ± 0.15 | 2.54 ± 0.17 | 0.20 ± 0.03 | 0.29 ± 0.04 | 0.28 ± 0.02 |
| +G2QDR w/ PT | 0.80 ± 0.04 | 0.63 ± 0.20 | 2.34 ± 0.21 | 0.19 ± 0.02 | 0.28 ± 0.03 | 0.33 ± 0.03 |
| HESS | 0.80 ± 0.04 | 0.74 ± 0.08 | 2.74 ± 0.15 | 0.77 ± 0.10 | 0.55 ± 0.12 | 0.53 ± 0.15 |
| +G4RL | 0.92 ± 0.02 | 0.84 ± 0.04 | 3.09 ± 0.21 | 0.74 ± 0.09 | 0.59 ± 0.12 | 0.56 ± 0.10 |
| +G2QDR w/o PT | 0.95 ± 0.03 | 0.80 ± 0.06 | 3.02 ± 0.13 | 0.82 ± 0.05 | 0.71 ± 0.13 | 0.58 ± 0.20 |
| +G2QDR w/ PT | 0.93 ± 0.04 | 0.78 ± 0.05 | 2.93 ± 0.24 | 0.85 ± 0.07 | 0.73 ± 0.11 | 0.68 ± 0.08 |
| HLPS | 0.78 ± 0.03 | 0.76 ± 0.11 | 2.96 ± 0.29 | 0.73 ± 0.09 | 0.71 ± 0.06 | 0.56 ± 0.17 |
| +G4RL | 0.94 ± 0.01 | 0.88 ± 0.04 | 3.11 ± 0.16 | 0.79 ± 0.02 | 0.64 ± 0.07 | 0.62 ± 0.07 |
| +G2QDR w/o PT | 0.95 ± 0.03 | 0.83 ± 0.03 | 3.31 ± 0.14 | 0.78 ± 0.04 | 0.77 ± 0.06 | 0.77 ± 0.05 |
| +G2QDR w/ PT | 0.90 ± 0.02 | 0.81 ± 0.07 | 3.03 ± 0.17 | 0.86 ± 0.04 | 0.79 ± 0.04 | 0.75 ± 0.07 |

- **a: High-level constraint rewards only**: Equation (8) is applied to the high-level rewards, while the low-level significance is set to $\alpha_l = 0$ in equation (9).

- **b: Low-level correction rewards only**: Equation (9) is applied to the low-level rewards, while the high-level significance is set to $\alpha_h = 0$ in equation (8).

- **c: Combined constraint rewards and correction rewards**: Equations (8) and (9) are applied simultaneously to the high-level and low-level reward structures, respectively.

- **d: Full G2QDR (with penalty terms)**: Both equations (8) and (9) are applied as in variant c, supplemented by the additional penalty terms defined in equation (10).

- **Baselines (HIRO/HRAC/HESS/HLPS)**: Standard implementations of the vanilla baseline methods for performance comparison.

As before, all data reported in this section are averaged over 10 independent runs and reported as the mean ± standard deviation.

Table 2 presents the results of the ablation study across all evaluated environments and algorithms. Overall, combining high-level constraints and low-level corrections (variant (c)) yields strong performance, often outperforming the use of either component in isolation. While the full G2QDR variant with penalty terms (d) further improves performance in several tasks, it does not uniformly dominate variant c, indicating that the effect of the penalty is environment-dependent.

In environments where Euclidean distance is a poor proxy for the actual transition distances, for example, in **AntMaze**, where walls separate states that are close in Euclidean space but far in terms of transition

Table 2: Ablation study of G2QDR variants. For each method, we report the performance of the original baseline together with its variants augmented by different types of auxiliary rewards (a–d). All results correspond to the final training episode. Within each method group, the best-performing variant is underlined.

| | AntMaze U-shape | AntMaze W-shape | AntGather | AntPush | AntFall | Pusher |
|---|---|---|---|---|---|---|
| HIRO | $0.72 \pm 0.09$ | $0.53 \pm 0.14$ | $1.47 \pm 0.18$ | $0.05 \pm 0.01$ | $0.23 \pm 0.04$ | $0.17 \pm 0.03$ |
| HIRO-a | $\underline{0.89 \pm 0.11}$ | $0.63 \pm 0.12$ | $1.82 \pm 0.14$ | $0.14 \pm 0.02$ | $0.27 \pm 0.02$ | $0.23 \pm 0.04$ |
| HIRO-b | $0.81 \pm 0.05$ | $0.58 \pm 0.15$ | $1.56 \pm 0.15$ | $0.04 \pm 0.01$ | $0.26 \pm 0.04$ | $0.14 \pm 0.02$ |
| HIRO-c | $0.87 \pm 0.08$ | $\underline{0.66 \pm 0.09}$ | $1.90 \pm 0.11$ | $0.12 \pm 0.01$ | $\underline{0.35 \pm 0.03}$ | $0.26 \pm 0.03$ |
| HIRO-d | $0.80 \pm 0.04$ | $0.59 \pm 0.07$ | $\underline{1.96 \pm 0.13}$ | $\underline{0.19 \pm 0.04}$ | $0.31 \pm 0.03$ | $\underline{0.30 \pm 0.03}$ |
| HRAC | $0.74 \pm 0.05$ | $0.59 \pm 0.27$ | $2.03 \pm 0.22$ | $0.10 \pm 0.01$ | $0.21 \pm 0.06$ | $0.24 \pm 0.04$ |
| HRAC-a | $0.81 \pm 0.04$ | $\underline{0.72 \pm 0.16}$ | $2.39 \pm 0.13$ | $\underline{0.22 \pm 0.02}$ | $0.19 \pm 0.05$ | $0.22 \pm 0.04$ |
| HRAC-b | $\underline{0.87 \pm 0.03}$ | $0.69 \pm 0.18$ | $2.27 \pm 0.26$ | $0.12 \pm 0.02$ | $0.24 \pm 0.03$ | $0.26 \pm 0.06$ |
| HRAC-c | $0.84 \pm 0.03$ | $0.70 \pm 0.15$ | $\underline{2.54 \pm 0.17}$ | $0.20 \pm 0.03$ | $\underline{0.29 \pm 0.04}$ | $0.28 \pm 0.02$ |
| HRAC-d | $0.80 \pm 0.04$ | $0.63 \pm 0.20$ | $2.34 \pm 0.21$ | $0.19 \pm 0.02$ | $0.28 \pm 0.03$ | $\underline{0.33 \pm 0.03}$ |
| HESS | $0.80 \pm 0.04$ | $0.74 \pm 0.08$ | $2.74 \pm 0.15$ | $0.77 \pm 0.10$ | $0.55 \pm 0.12$ | $0.53 \pm 0.15$ |
| HESS-a | $0.87 \pm 0.03$ | $0.76 \pm 0.05$ | $2.88 \pm 0.27$ | $0.74 \pm 0.12$ | $0.68 \pm 0.16$ | $0.60 \pm 0.11$ |
| HESS-b | $0.92 \pm 0.02$ | $\underline{0.81 \pm 0.07}$ | $2.85 \pm 0.10$ | $0.79 \pm 0.09$ | $0.66 \pm 0.08$ | $0.47 \pm 0.22$ |
| HESS-c | $\underline{0.95 \pm 0.03}$ | $0.80 \pm 0.06$ | $\underline{3.02 \pm 0.13}$ | $0.82 \pm 0.05$ | $0.71 \pm 0.13$ | $0.58 \pm 0.20$ |
| HESS-d | $0.93 \pm 0.04$ | $0.78 \pm 0.05$ | $2.93 \pm 0.24$ | $\underline{0.85 \pm 0.07}$ | $\underline{0.73 \pm 0.11}$ | $\underline{0.68 \pm 0.08}$ |
| HLPS | $0.78 \pm 0.03$ | $0.76 \pm 0.11$ | $2.96 \pm 0.29$ | $0.73 \pm 0.09$ | $0.71 \pm 0.06$ | $0.56 \pm 0.13$ |
| HLPS-a | $0.91 \pm 0.02$ | $0.79 \pm 0.07$ | $\underline{3.36 \pm 0.20}$ | $0.79 \pm 0.08$ | $0.77 \pm 0.05$ | $0.66 \pm 0.09$ |
| HLPS-b | $0.88 \pm 0.03$ | $0.81 \pm 0.05$ | $3.20 \pm 0.24$ | $0.70 \pm 0.06$ | $0.69 \pm 0.08$ | $0.58 \pm 0.11$ |
| HLPS-c | $\underline{0.95 \pm 0.03}$ | $\underline{0.83 \pm 0.03}$ | $3.31 \pm 0.14$ | $0.78 \pm 0.04$ | $0.77 \pm 0.06$ | $\underline{0.77 \pm 0.05}$ |
| HLPS-d | $0.90 \pm 0.02$ | $0.81 \pm 0.07$ | $3.03 \pm 0.17$ | $\underline{0.86 \pm 0.04}$ | $\underline{0.79 \pm 0.04}$ | $0.75 \pm 0.07$ |

distance, incorporating the low-level correction term leads to clear and consistent gains, highlighting its role in improving local policy accuracy. The high-level constraint term also contributes to performance improvements by guiding the policy toward more feasible subgoals, although its effect is generally more pronounced when combined with low-level corrections.

The penalty term exhibits mixed behavior across tasks. It tends to improve performance in environments with asymmetric or partially irreversible dynamics, such as Pusher and AntPush, where it discourages transitions to states that are easy to reach but difficult to recover from. However, in more symmetric environments, the same penalty can restrict exploration and slightly degrade performance. These results suggest that, while the full G2QDR formulation is beneficial in certain settings, the combination of constraints and corrections alone provides strong performance overall.

### 4.3.2 Balancing between time and performance

Several aspects of our method involve a tradeoff between performance and computational cost:

- **State sampling frequency:** Lowering the sampling frequency of candidate states for node features reduces comparisons during graph updates but may weaken the graph's representativeness.

- **Training data selection:** Training $\mathcal{C}_\theta$ on a subset of node pairs and edges speeds up training, at the potential cost of accuracy.

We first vary the sampling frequency of candidate states used as node features, considering intervals of 1, 5, and 10 steps in HLPS for **AntPush** and **Pusher**.

All curves reported in Section 4 represent averages over 10 independent runs, with standard deviations shown as shaded regions. For clarity, all results are uniformly smoothed for visualization.

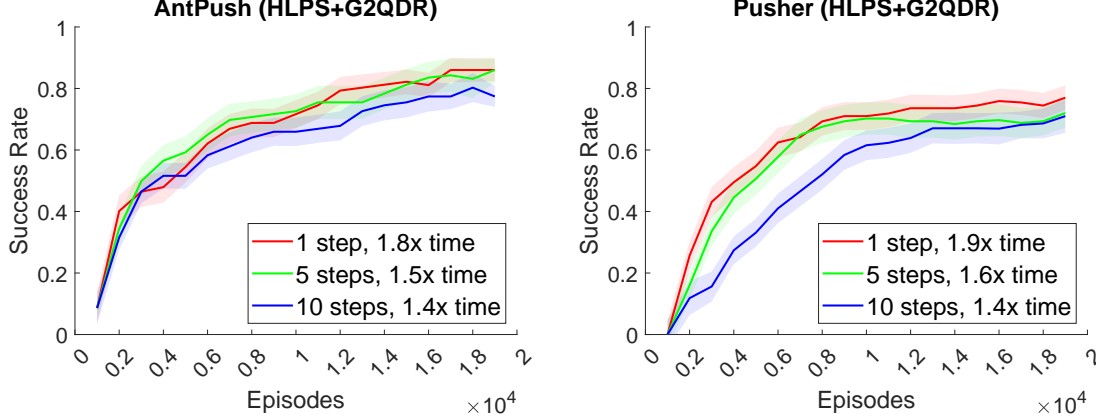

Figure 2: Success rates on (a) AntPush and (b) Pusher using HLPS+G2QDR. The number of steps in the legend denotes the chosen state sampling frequency, and the timescale is reported relative to the vanilla HLPS algorithm. Increasing the sampling interval substantially reduces computation time, with only minor degradation in success rates across both tasks.

As shown in Figure 2, increasing the sampling interval substantially reduces computation time by reducing the number of graph interactions, while causing only minor degradation in success rates across both tasks.

Next, we vary the proportion of training data used for the state connectivity model, evaluating sizes of 50%, 75%, and 100% of all available training samples in the graph. The results in Figure 3 show that reducing the amount of training data yields only modest improvements in computational efficiency. In particular, decreasing the dataset size slightly shortens computation time, but the gains are limited. At the same time, the success rates on both AntPush and Pusher remain largely stable, with only minor degradation when using 75% of the available training samples. This suggests that G2QDR is relatively robust to moderate data reductions and does not rely heavily on the full dataset to achieve strong performance.

These findings suggest that the primary computational bottleneck of G2QDR lies in the graph construction and node comparison procedures described in Section 3.1.1, rather than in the model training stage. Consequently, adjusting the sampling frequency emerges as an effective way to reduce computational cost, while largely preserving the performance gains provided by G2QDR integration.

## 4.4 Experimental analysis of dense signal schedule variations

This section examines how different dense reward schedules influence the performance of G2QDR. We introduce several variants of dense signal schedules, illustrated in Figure 4 (b), and present the corresponding success rate curves for **Pusher** using HLPS+G2QDR in Figure 4 (a).

From Figure 4, several key observations can be drawn. An immediate full reward setting ($\lambda = 1$) leads to slower convergence compared to other configurations, suggesting that instability in the state graph and the state connectivity model adversely affects both efficiency and convergence speed of the backbone method. Moreover, keeping $\lambda$ high throughout later episodes results in inferior performance relative to variants that apply late-stage annealing. In contrast, a well-designed schedule for $\lambda$ enables the architecture to achieve its full potential.

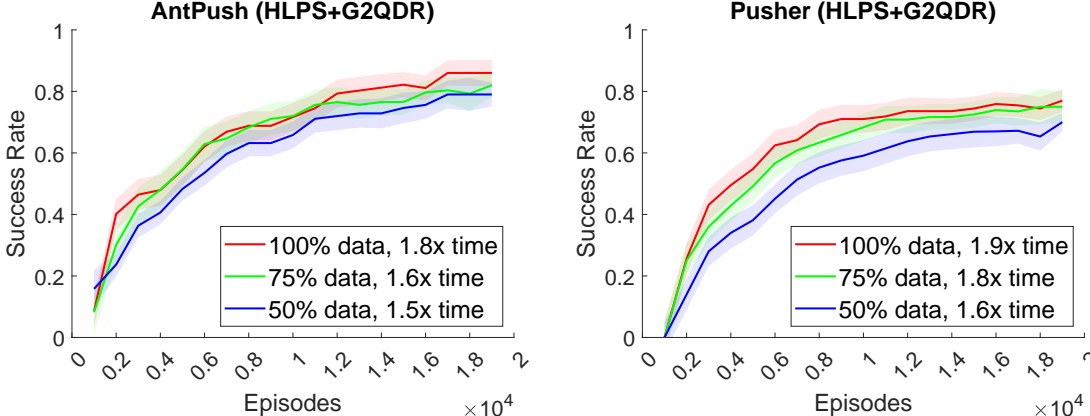

Figure 3: Success rate on (a) AntPush and (b) Pusher using HLPS+G2QDR. The percentages in the legend indicate the proportion of training samples used relative to the total number of edges in the graph. The timescale is computed with respect to the vanilla HLPS algorithm. Reducing the amount of data slightly decreases computation time; using 75% of the data results in only minor degradation in success rates across both AntPush and Pusher tasks.

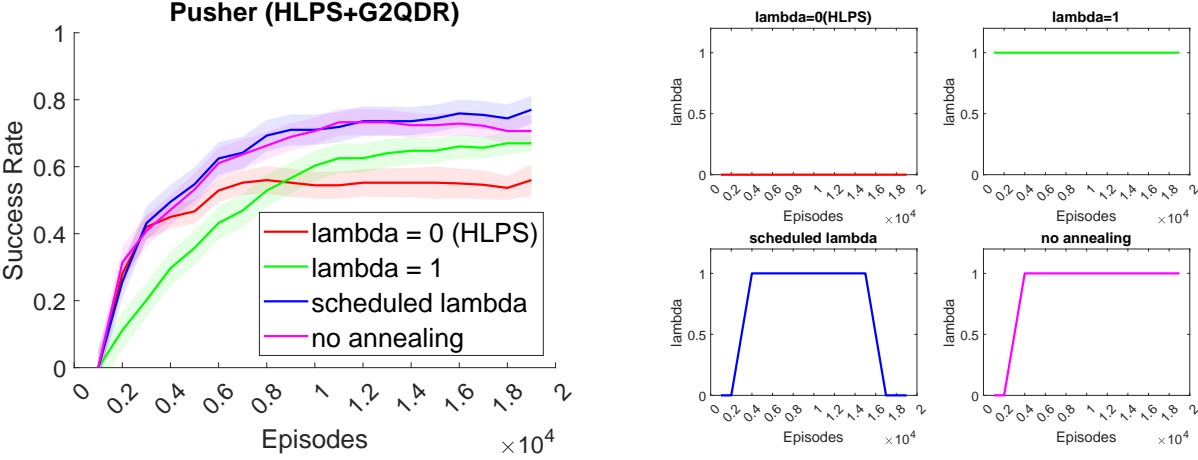

Figure 4: Ablation study of dense signal-strength scheduling. **Left:** Training curves under different $\lambda$ schedules. **Right:** Evolution of the scheduling parameter $\lambda$ across episodes.

## 5   Conclusion

In this paper, we present a unified framework for graph-based GCHRL that incrementally constructs a directed state graph during exploration, learns a state connectivity model to estimate transition feasibility and evaluate newly encountered states, and leverages connectivity-based auxiliary rewards to enhance learning efficiency across multiple levels of agents.

Building upon the G4RL framework (Zhang et al., 2025), our approach improves and extends its capabilities by explicitly modeling asymmetric state transitions, making it well-suited for asymmetric environments where transition dynamics are inherently directional and non-reversible. This enhancement allows the architecture to better capture realistic environment structures that are often overlooked by symmetric assumptions.

The proposed framework is broadly compatible with existing GCHRL methods, enabling seamless integration with a wide range of prior approaches. Empirical evaluations across diverse environments demonstrate consistent improvements over baseline methods, highlighting the effectiveness and robustness of our approach.

Overall, this work highlights the importance of modeling directional structure in state spaces for goal-conditioned hierarchical reinforcement learning. By translating state connectivity into auxiliary rewards, the proposed framework offers a flexible and scalable approach for tackling complex environments.

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

# A    Limitations and Future Work

Despite the promising advantages demonstrated by G2QDR in hierarchical reinforcement learning (HRL) tasks, several limitations remain. First, its performance is highly sensitive to a number of key hyperparameters, including $\epsilon_d$, the significance-related hyperparameters ($\alpha$), $p$, and $W$. Obtaining strong and stable performance across diverse environments therefore requires careful and often time-consuming manual tuning, which may limit the scalability and practical applicability of the method.

Second, the graph is constructed dynamically during exploration and is consequently influenced by the behavior policy of the underlying backbone method. Such policy-dependent graph construction introduces exploration bias, potentially resulting in an incomplete or unrepresentative approximation of the environment structure. As a consequence, the dense rewards induced from the learned graph may inherit this bias, which could affect the quality of guidance provided to the agent.

Third, the environments considered in our experiments are predominantly continuous and characterized by high-dimensional state representations. In these settings, the ground-truth state connectivity structure and the true transition distances between states are generally unavailable. Therefore, directly evaluating the quality of the constructed graph against an optimal or reference graph is infeasible. Instead, the effectiveness of the learned graph is evaluated indirectly through its impact on downstream task performance.

Finally, the introduced auxiliary reward is not potential-based, indicating that it alters the optimization objective and changes the optimal policy. Although we introduce a dense signal scheduling strategy to mitigate this issue, the auxiliary reward may still influence exploration bias and the replay buffer, which could potentially affect the performance of our method.

In future work, we aim to address these limitations by developing adaptive mechanisms that can automatically select or adjust key hyperparameters according to environmental characteristics and learning dynamics. Such approaches could improve the robustness and generalization capability of G2QDR while reducing the reliance on manual tuning. Furthermore, we plan to enhance the scalability of G2QDR by improving its computational efficiency and evaluating its performance in larger and more complex environments, particularly those with higher-dimensional state and action spaces. Finally, we will investigate more principled approaches for evaluating the quality of learned graphs and reducing the impact of exploration bias during graph construction, enabling more reliable and generalizable dense reward generation.

# B Implementation details

## B.1 Environment details

**AntMaze-U Shape** This environment is part of the Gymnasium-Robotics benchmark suite. The environment has a size of $(24 \times 24)$. Both the state and action spaces are continuous, with state and action dimensions of 31 and 8, respectively. A reward of 1 is issued only when the agent reaches a position within a Euclidean distance of 1 from the goal; otherwise, the reward is 0. During evaluation, the goal is fixed at $(0, 16)$, and an episode is considered successful if the agent finishes within a Euclidean distance of 1 from the goal.

**AntMaze-W Shape** This environment is a variant of AntMaze-U Shape with an enlarged $(32 \times 32)$ maze. The state and action spaces are identical to those of AntMaze-U Shape. As in the previous environment, the agent receives a reward of 1 only when it reaches within a Euclidean distance of 1 from the goal and 0 otherwise. The goal is fixed at $(2, 16)$.

**AntGather** This environment is introduced by Duan et al. (2016). The environment size is $(20 \times 20)$, and both the state and action spaces are continuous. The objective is to collect apples while avoiding bombs. The agent receives a reward of $(+1)$ for each apple collected and a penalty of $(-1)$ for each bomb collected. Apples and bombs are randomly distributed throughout the environment.

**AntPush** This environment has a size of $(20 \times 20)$ with continuous state and action spaces. It is a challenging manipulation task requiring both navigation and object interaction. To reach the goal, the agent must first move left, then upward, and finally push a movable block to the right. Incorrectly pushing the block may permanently obstruct the path to the goal.

**AntFall** This environment also has a size of $(20 \times 20)$. The task requires the agent to cross a chasm by pushing a block into it to form a bridge. If the agent falls into the chasm, recovery is impossible, and the episode effectively terminates.

**Pusher** In this environment, the agent controls a 7-DoF robotic manipulator by applying continuous torques to its joints. The state space consists of the joint positions and velocities, together with the positions of the manipulated object and the target. The objective is to push the object to a designated target location.

For these environments, no reward information is communicated between the high-level and low-level agents. Consequently, no aggregation of low-level rewards is performed. Since the environment provides only a terminal sparse reward, each high-level transition is assigned the auxiliary reward generated by our framework, in addition to the terminal environment reward.

## B.2 Network architecture details

Our network architecture for the HRL agents follows prior work (Nachum et al., 2018b; Zhang et al., 2022; Li et al., 2022; Wang et al., 2024). For HIRO, HRAC, HESS, and HLPS, both the high-level and low-level policies are implemented using TD3. The actor and critic networks in TD3 each consist of hidden layers with size 300.

For the state connectivity model, we employ a four-layer fully connected network with a hidden dimension of 128 and ReLU activation.

We use Adam as the optimizer for the actor and critic networks, as well as for the state connectivity model (Kingma & Ba, 2014).

## C Hyperparameters

In this section we list the values of hyperparameters used in our experiments.

| Hyperparameters | Values |
|---|---|
| High-level agent | |
| Actor learning rate | 0.0001 |
| Critic learning rate | 0.001 |
| Batch size | 128 |
| Discount factor $\gamma$ | 0.99 |
| Policy update frequency | 1 |
| High-level action frequency | 10 |
| Replay buffer size | 20000 |
| Exploration strategy | Gaussian($\sigma = 1$) |
| Low-level agent | |
| Actor learning rate | 0.0001 |
| Critic learning rate | 0.001 |
| Batch size | 128 |
| Discount factor $\gamma$ | 0.99 |
| Policy update frequency | 1 |
| Replay buffer size | 20000 |
| Exploration strategy | Gaussian($\sigma = 1$) |

Table 3: Hyperparameters used in high- and low-level TD3 agents.

| Hyperparameters | Values |
|---|---|
| Number of nodes $N$ | 200 |
| Batch size | 128 |
| Optimizer learning rate | 0.0001 |
| $\epsilon_d$ | 0.5 for AntMaze/1 for others |
| $W$ | 5 |
| $p$ | 2 |
| $\alpha_h$ | 0.005 |
| $\alpha_l$ | 0.005 |
| $\alpha_{hp}$ | 0.01 |
| $\alpha_{lp}$ | 0.01 |
| $n_1$ | 2000 |
| $n_2$ | 4000 |
| $n_3$ | 15000 |
| $n_4$ | 17000 |

Table 4: Hyperparameters used in G2QDR.

The state representation function $\phi(\cdot)$ is a feature selection function that maps the raw state to a reduced representation by retaining only location-related features, such as the positions of the agent, objects, and

designated targets (or goals). All other state variables, including arm angular velocities, joint angles, and other kinematic features, are discarded.

The hyperparameters $n_1$ through $n_4$ are chosen such that $n_1$ and $n_2$ fall within the early stage of training, while $n_3$ and $n_4$ correspond to the late stage. All experiments are conducted using this single choice of $n_1$ to $n_4$, and alternative configurations were not explored.

## D  Additional experimental evaluations

### D.1  Measure of spatial asymmetry

Although measures such as forward/reverse reachability gaps would provide a more precise characterization, they are difficult to estimate reliably in continuous control environments with high-dimensional state spaces. Instead, for **AntMaze**, **AntGather**, **AntFall**, **AntPush**, and **Pusher**, we introduce rule-based criteria that approximate transition asymmetry induced by environment structure and constraints. The criteria are defined as follows.

**AntMaze (U-shape/W-shape)**  All transitions between arbitrary pairs of states are considered symmetric, as these environments contain no irreversible obstacles or traps.

**AntGather**  We define an unrecoverable region adjacent to each wall, within which an object cannot be retrieved because the agent is unable to pull it away from the wall. Consequently, transitions between states in which a given object moves into or out of this region are considered asymmetric.

**AntFall**  We determine whether the agent or the block is located inside the chasm. If two states differ in this status for either the agent or the block (i.e., inside versus outside the chasm), the transitions between them are considered asymmetric.

**AntPush**  Similar to **AntGather**, we define an unrecoverable region adjacent to each wall, within which the block cannot be retrieved because the agent is unable to pull it away from the wall. Consequently, transitions between states in which the block moves into or out of this region are considered asymmetric.

**Pusher**  We partition the state space into two categories depending on whether the object lies within the robotic arm's reachable workspace. Transitions between states belonging to different categories are considered asymmetric.

Although these rule-based criteria do not fully characterize all kinds of transition asymmetry for state pairs, they provide a tractable approximation for quantifying spatial asymmetry induced by environmental geometry and constraints.

We estimate an asymmetry score for each environment by uniformly sampling 10,000 state pairs from the valid state space and computing the fraction classified as asymmetric under the proposed criteria. This score estimates the probability that a randomly sampled state pair is classified as asymmetric under the proposed criteria, providing a tractable proxy for structural transition asymmetry. The scores are reported in Table 5.

Table 5: **Asymmetry scores across environments.** We estimate the asymmetry score by uniformly sampling 10,000 state pairs and computing the fraction classified as asymmetric under the proposed rule-based criteria.

| | AntMaze U-shape | AntMaze W-shape | AntGather | AntPush | AntFall | Pusher |
|---|---|---|---|---|---|---|
| Asymmetry score | 0.00 | 0.00 | 0.42 | 0.56 | 0.48 | 0.39 |

The estimated asymmetry scores are consistent with the qualitative discussion in Section 4.2: the directed G2QDR is more likely to outperform its undirected counterpart, G4RL, on **AntGather**, **AntPush**, **AntFall**,

and **Pusher**, supporting our characterization of these environments as exhibiting greater structural asymmetry.

## D.2  Direct evaluation of the state connectivity model

Based on the rule-based asymmetry criteria defined in Section D.1, we evaluate the absolute asymmetry gap $|\mathcal{C}_\theta(\psi(s_i, s_j)) - \mathcal{C}_\theta(\psi(s_j, s_i))|$ of the state connectivity model $\mathcal{C}_\theta(\cdot)$ using symmetric and asymmetric state pairs sampled from the valid state space. The difference in this quantity between symmetric and asymmetric pairs can be interpreted as an indicator of whether the model captures underlying directional structure in the environments. Figure 5 reports the average absolute asymmetry gap for each group at different stages of training. The underlying training backbone is **HLPS**.

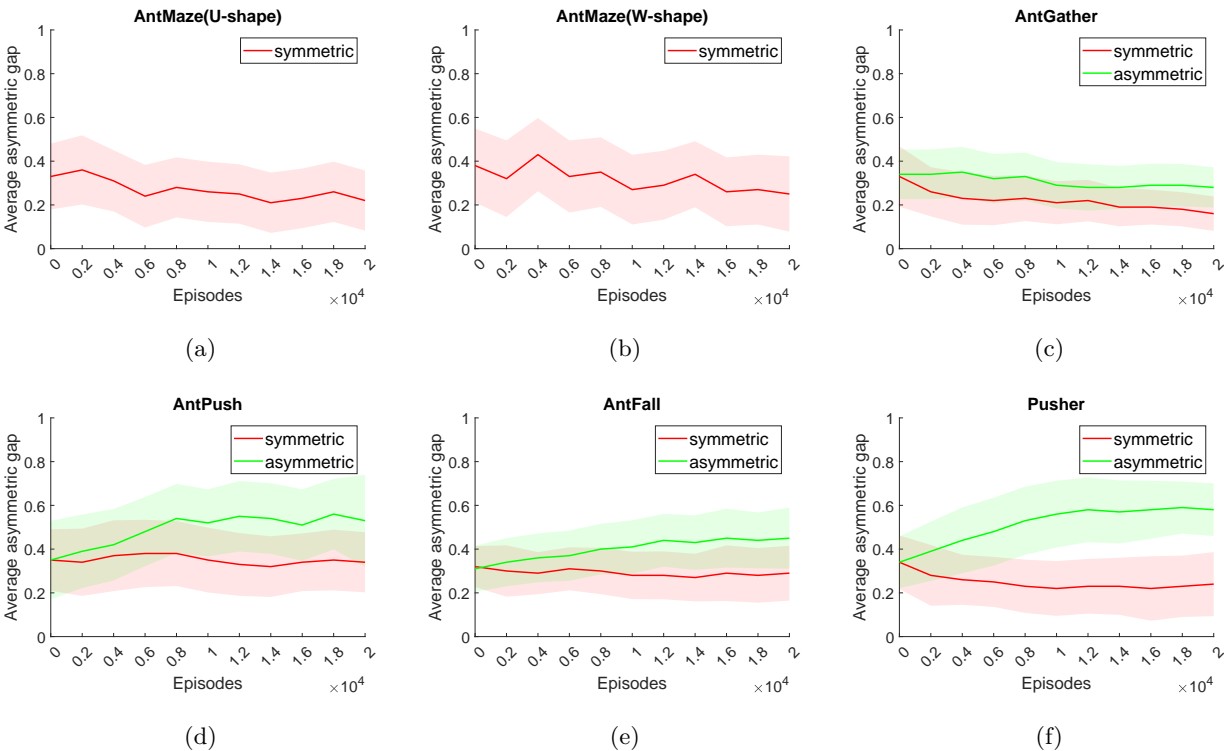

Figure 5: **Symmetry gap in different environments.** We report the average absolute asymmetry gaps for sampled symmetric and asymmetric state pairs. The red curves show the evolution of the averaged absolute asymmetry gap under the state connectivity model $\mathcal{C}_\theta(\cdot)$ for symmetric state pairs, while the green curves show the corresponding values for asymmetric state pairs. Note that the **AntMaze** environments contain only symmetric state pairs under our rule-based criteria.

The overall trend shown in the figures is that the separation between the two curves gradually increases across environments. This suggests that the model is indeed learning to distinguish asymmetric state pairs from symmetric ones by assigning larger asymmetry gaps to the former.

**Direct diagnosis of $\mathcal{C}_\theta(\cdot)$.**  To directly evaluate whether the learned state connectivity scores reflect transition feasibility, we perform a diagnostic analysis on **AntMaze**. Since the environment contains only a single movable agent and its dynamics are nearly deterministic and reversible, graph shortest-path distance provides a reliable proxy for empirical reachability between states.

We discretize the workspace into a regular square grid with spacing 0.1, such that the coordinates of each grid point are multiples of 0.1. Adjacent valid grid points are connected to form a square lattice graph. For two arbitrary states $s_i$ and $s_j$, we map them to their nearest grid vertices $v_i$ and $v_j$, respectively, and compute

the shortest-path distance on the grid graph, denoted by $d_g(v_i, v_j)$. Because transitions between nearby states generally require fewer actions and have higher rollout success under the deterministic dynamics of AntMaze, this graph-based distance serves as a practical approximation of transition feasibility.

To evaluate whether the learned connectivity model preserves this notion of feasibility, we uniformly sample four states from the valid state space, denoted by $s_1$, $s_2$, $s_3$, and $s_4$, and compare the ordering induced by the connectivity scores with the ordering induced by the graph-based distances. Specifically, a sampled quadruple is considered monotonicity preserving if

$$(\mathcal{C}_\theta(\psi(s_1, s_2)) > \mathcal{C}_\theta(\psi(s_3, s_4))) \iff (d_g(v_1, v_2) < d_g(v_3, v_4)). \tag{12}$$

We uniformly sample a large number of quadruples and compute the fraction that preserves monotonicity. A higher monotonicity-preservation rate indicates that the learned connectivity scores rank state pairs more consistently with their underlying transition feasibility. As shown in Figure 6, this metric generally improves throughout training on both AntMaze(U-shape) and AntMaze(W-shape), demonstrating that $\mathcal{C}_\theta(\cdot)$ becomes increasingly aligned with an independent reachability proxy rather than merely exhibiting directional score differences.

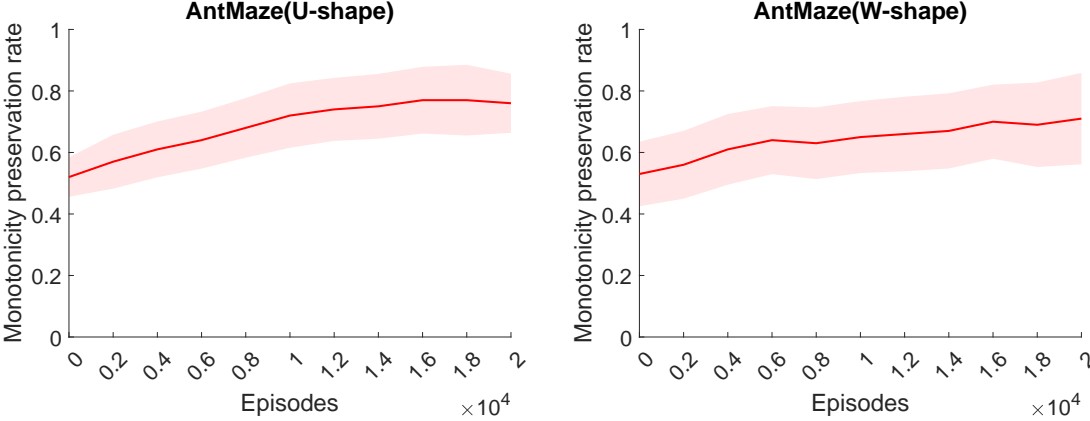

Figure 6: Monotonicity preservation rate on (a) AntMaze U-shape and (b) AntMaze W-shape using HLPS+G2QDR. The monotonicity-preservation rate generally increases during training, indicating that the learned connectivity model becomes increasingly aligned with the graph-based transition-feasibility proxy.

### D.3 Ablation study on hyperparameters

In this section, we present an ablation study to evaluate the influence of several key hyperparameters on the performance of the proposed framework on **HESS** and **HLPS**. Specifically, we investigate the effects of the window size $W$, the decay exponent $p$, the novelty threshold $\epsilon_d$, and the reward weights $\alpha$. Unless otherwise specified, all hyperparameters are fixed to the values reported in Section C, with only the hyperparameter under investigation varied in each experiment.

From Table 6, we observe that increasing the window size $W$ improves the final performance up to a certain point. Beyond this point, however, the gains quickly plateau, and further increasing $W$ yields only marginal or no improvement while incurring substantially higher computational overhead.

Table 7 shows that our method performs well when $p = 1$ or $p = 2$. However, increasing $p$ causes the connection weights to decay much more rapidly over time, making the edge weights between temporally distant states negligible. Consequently, long-range temporal dependencies contribute little to the graph, causing the model to behave similarly to the $W = 1$ setting.

Table 8 shows that in general $\epsilon_d = 0.5$ or $1$ yields the best overall performance. When $\epsilon_d$ is too small, the graph structure changes too frequently due to the continual insertion and removal of nodes, preventing the

Table 6: Ablation study on the window size $W$

|  | AntMaze U-shape | AntMaze W-shape | AntGather | AntPush | AntFall | Pusher |
|---|---|---|---|---|---|---|
| **HESS** | | | | | | |
| $W = 1$ | $0.83 \pm 0.03$ | $0.70 \pm 0.06$ | $2.76 \pm 0.19$ | $0.72 \pm 0.09$ | $0.61 \pm 0.14$ | $0.65 \pm 0.13$ |
| $W = 5$ | $0.93 \pm 0.04$ | $0.78 \pm 0.05$ | $2.93 \pm 0.24$ | $0.85 \pm 0.07$ | $0.73 \pm 0.11$ | $0.68 \pm 0.08$ |
| $W = 10$ | $0.95 \pm 0.02$ | $0.81 \pm 0.04$ | $2.87 \pm 0.20$ | $0.87 \pm 0.09$ | $0.72 \pm 0.09$ | $0.70 \pm 0.10$ |
| $W = 20$ | $0.95 \pm 0.02$ | $0.80 \pm 0.04$ | $2.95 \pm 0.17$ | $0.85 \pm 0.08$ | $0.70 \pm 0.12$ | $0.72 \pm 0.09$ |
| **HLPS** | | | | | | |
| $W = 1$ | $0.86 \pm 0.03$ | $0.85 \pm 0.04$ | $3.14 \pm 0.20$ | $0.80 \pm 0.06$ | $0.73 \pm 0.05$ | $0.65 \pm 0.12$ |
| $W = 5$ | $0.90 \pm 0.02$ | $0.81 \pm 0.07$ | $3.03 \pm 0.17$ | $0.86 \pm 0.04$ | $0.79 \pm 0.04$ | $0.75 \pm 0.07$ |
| $W = 10$ | $0.89 \pm 0.04$ | $0.84 \pm 0.05$ | $3.17 \pm 0.24$ | $0.83 \pm 0.05$ | $0.79 \pm 0.05$ | $0.76 \pm 0.10$ |
| $W = 20$ | $0.92 \pm 0.03$ | $0.84 \pm 0.08$ | $3.16 \pm 0.22$ | $0.86 \pm 0.05$ | $0.80 \pm 0.06$ | $0.73 \pm 0.08$ |

Table 7: Ablation study on the decay exponent $p$

|  | AntMaze U-shape | AntMaze W-shape | AntGather | AntPush | AntFall | Pusher |
|---|---|---|---|---|---|---|
| **HESS** | | | | | | |
| $p = 1$ | $0.92 \pm 0.02$ | $0.81 \pm 0.06$ | $3.06 \pm 0.13$ | $0.84 \pm 0.08$ | $0.77 \pm 0.08$ | $0.62 \pm 0.09$ |
| $p = 2$ | $0.93 \pm 0.04$ | $0.78 \pm 0.05$ | $2.93 \pm 0.24$ | $0.85 \pm 0.07$ | $0.73 \pm 0.11$ | $0.68 \pm 0.08$ |
| $p = 5$ | $0.84 \pm 0.03$ | $0.73 \pm 0.05$ | $2.80 \pm 0.15$ | $0.78 \pm 0.05$ | $0.67 \pm 0.09$ | $0.64 \pm 0.12$ |
| **HLPS** | | | | | | |
| $p = 1$ | $0.92 \pm 0.03$ | $0.85 \pm 0.06$ | $3.21 \pm 0.28$ | $0.82 \pm 0.07$ | $0.77 \pm 0.10$ | $0.78 \pm 0.09$ |
| $p = 2$ | $0.90 \pm 0.02$ | $0.81 \pm 0.07$ | $3.03 \pm 0.17$ | $0.86 \pm 0.04$ | $0.79 \pm 0.04$ | $0.75 \pm 0.07$ |
| $p = 5$ | $0.85 \pm 0.05$ | $0.76 \pm 0.05$ | $3.00 \pm 0.19$ | $0.79 \pm 0.07$ | $0.74 \pm 0.07$ | $0.70 \pm 0.10$ |

edge weights from remaining well synchronized. Conversely, when $\epsilon_d$ is too large, each node may aggregate too many distinct states, resulting in overly coarse graph representations and biased connectivity scores between those states.

Table 9 shows that when $\alpha$ is small, the contribution of the additional reward signal becomes limited, causing the method to approach the behavior of the backbone method. In contrast, a large $\alpha$ may cause the auxiliary reward to dominate the environmental reward, shifting the optimization objective and potentially degrading performance when the two objectives are misaligned.

Table 8: Ablation study on the novelty threshold $\epsilon_d$

| | AntMaze U-shape | AntMaze W-shape | AntGather | AntPush | AntFall | Pusher |
|---|---|---|---|---|---|---|
| HESS | | | | | | |
| $\epsilon_d = 0.1$ | $0.85 \pm 0.06$ | $0.81 \pm 0.07$ | $2.82 \pm 0.17$ | $0.76 \pm 0.08$ | $0.66 \pm 0.15$ | $0.61 \pm 0.08$ |
| $\epsilon_d = 0.5$ | $0.93 \pm 0.04$ | $0.78 \pm 0.05$ | $3.04 \pm 0.14$ | $0.82 \pm 0.04$ | $0.75 \pm 0.06$ | $0.56 \pm 0.10$ |
| $\epsilon_d = 1$ | $0.91 \pm 0.03$ | $0.79 \pm 0.04$ | $2.93 \pm 0.24$ | $0.85 \pm 0.07$ | $0.73 \pm 0.11$ | $0.68 \pm 0.08$ |
| $\epsilon_d = 5$ | $0.80 \pm 0.07$ | $0.71 \pm 0.13$ | $3.13 \pm 0.29$ | $0.71 \pm 0.13$ | $0.69 \pm 0.19$ | $0.63 \pm 0.05$ |
| HLPS | | | | | | |
| $\epsilon_d = 0.1$ | $0.76 \pm 0.10$ | $0.78 \pm 0.04$ | $2.62 \pm 0.14$ | $0.84 \pm 0.06$ | $0.61 \pm 0.15$ | $0.69 \pm 0.11$ |
| $\epsilon_d = 0.5$ | $0.90 \pm 0.02$ | $0.81 \pm 0.07$ | $2.96 \pm 0.22$ | $0.82 \pm 0.06$ | $0.72 \pm 0.09$ | $0.76 \pm 0.05$ |
| $\epsilon_d = 1$ | $0.88 \pm 0.03$ | $0.85 \pm 0.03$ | $3.03 \pm 0.17$ | $0.86 \pm 0.04$ | $0.79 \pm 0.04$ | $0.75 \pm 0.07$ |
| $\epsilon_d = 5$ | $0.81 \pm 0.05$ | $0.73 \pm 0.08$ | $3.08 \pm 0.16$ | $0.75 \pm 0.09$ | $0.53 \pm 0.12$ | $0.72 \pm 0.13$ |

Table 9: Ablation study on the reward weights $\alpha_h$ and $\alpha_l$

| | AntMaze U-shape | AntMaze W-shape | AntGather | AntPush | AntFall | Pusher |
|---|---|---|---|---|---|---|
| HESS | | | | | | |
| $\alpha_h, \alpha_l = 0.001$ | $0.82 \pm 0.04$ | $0.79 \pm 0.07$ | $2.77 \pm 0.36$ | $0.76 \pm 0.06$ | $0.60 \pm 0.16$ | $0.50 \pm 0.13$ |
| $\alpha_h, \alpha_l = 0.005$ | $0.93 \pm 0.04$ | $0.78 \pm 0.05$ | $2.93 \pm 0.24$ | $0.85 \pm 0.07$ | $0.73 \pm 0.11$ | $0.68 \pm 0.08$ |
| $\alpha_h, \alpha_l = 0.01$ | $0.91 \pm 0.02$ | $0.81 \pm 0.06$ | $3.19 \pm 0.29$ | $0.83 \pm 0.03$ | $0.67 \pm 0.15$ | $0.65 \pm 0.07$ |
| HLPS | | | | | | |
| $\alpha_h, \alpha_l = 0.001$ | $0.77 \pm 0.05$ | $0.79 \pm 0.06$ | $2.87 \pm 0.25$ | $0.72 \pm 0.06$ | $0.74 \pm 0.05$ | $0.61 \pm 0.08$ |
| $\alpha_h, \alpha_l = 0.005$ | $0.90 \pm 0.02$ | $0.81 \pm 0.07$ | $3.03 \pm 0.17$ | $0.86 \pm 0.04$ | $0.79 \pm 0.04$ | $0.75 \pm 0.07$ |
| $\alpha_h, \alpha_l = 0.01$ | $0.92 \pm 0.02$ | $0.75 \pm 0.08$ | $3.22 \pm 0.20$ | $0.75 \pm 0.09$ | $0.68 \pm 0.15$ | $0.73 \pm 0.13$ |

