# OpenReview forum: "From Connectivity to Rewards: Dense Reward Learning with Directed State Graphs"
_TMLR — Under review for TMLR_

### Review · Reviewer_1sVS · 2026-06-25

**Summary Of Contributions:**

This paper introduces a state connectivity model designed to predict pairwise state connectivity strength in asymmetric environments to encode connectivity and state-accessibility information. Empirical results indicate that the proposed method demonstrates strong performance in most environments with minimal computational overhead. However, the core contributions of this paper do not seem to be clearly and effectively presented. Furthermore, the Method section merely describes a relatively complex implementation procedure. It is recommended to include a framework diagram to facilitate readers' understanding.

**Audience:**

No

**Audience Explanation:**

See the evaluation for accurate, convincing, and clear evidence.

**Claims And Evidence:**

No

**Claims Explanation:**

1. The core contributions of this paper do not seem to be clearly and effectively presented. Furthermore, the Method section merely describes a relatively complex implementation procedure. It is recommended to include a framework diagram to facilitate readers' understanding.
2. Equation (4) appears to be incorrect. The authors are kindly requested to double-check it carefully or provide a detailed explanation.
3. The overall structure of this paper has some issues. For instance, Section 3.5 should not be placed in the Method section; additionally, the algorithm procedure in Section 3.6 should be described in detail in the main text, rather than being entirely relegated to the appendix.
4. The claims regarding the experimental results in this paper appear to be overstated. For example, the performance of the HLPS+G2QDR algorithm in Table 1 is not consistently the best, particularly in the AntMaze W-shape environment. However, the paper describes it as "incorporating G2QDR into the base GCHRL methods consistently improves performance across environments."
5. There are numerous grammatical errors, typos, and formatting inconsistencies in this paper. The authors are advised to proofread the manuscript carefully. For example, the citation formats are inconsistent, some sentences lack ending punctuation, and there are obvious errors in the algorithm descriptions.

**Requested Changes:**

See the evaluation for accurate, convincing, and clear evidence.

---

> ### Author Response · Authors · 2026-07-02
> **Reply to reviewer 1sVS**
>
> Thank you for the time and effort you devoted to reviewing our manuscript and for your valuable comments and suggestions. We have carefully revised the manuscript in accordance with your recommendations. A point-by-point response detailing our revisions is provided below.
>
> 1. **The core contributions of this paper do not seem to be clearly and effectively presented. Furthermore, the Method section merely describes a relatively complex implementation procedure. It is recommended to include a framework diagram to facilitate readers' understanding.**
>
> Response: To better highlight the core contributions of our work and improve the readability of the Method section, we have added a framework diagram, along with a brief overview of the proposed framework, at the beginning of Section 3. We believe this addition provides readers with a clearer understanding of the overall methodology and the main contributions of our work before the detailed implementation is presented.
>
> 2. **Equation (4) appears to be incorrect. The authors are kindly requested to double-check it carefully or provide a detailed explanation.**
>
> Response: We carefully reviewed Equation (4) and confirmed its correctness. To improve its clarity and make it easier for readers to understand, we have added a more detailed explanation of the equation and its components in the revised manuscript.
>
> 3. **The overall structure of this paper has some issues. For instance, Section 3.5 should not be placed in the Method section; additionally, the algorithm procedure in Section 3.6 should be described in detail in the main text, rather than being entirely relegated to the appendix.**
>
> Response: We have revised the manuscript's organization accordingly. Specifically, the original Section 3.5 has been moved to the Experiments section, as it is more appropriate there. In addition, the algorithm previously presented in the appendix has been incorporated into the main text.
>
> 4. **The claims regarding the experimental results in this paper appear to be overstated. For example, the performance of the HLPS+G2QDR algorithm in Table 1 is not consistently the best, particularly in the AntMaze W-shape environment. However, the paper describes it as "incorporating G2QDR into the base GCHRL methods consistently improves performance across environments."**
>
> Response: In the revised manuscript, we have softened the relevant claims by using more precise expressions such as "in general" and "in most cases" to better reflect the experimental results. These revisions ensure that our conclusions are more accurately aligned with the empirical evidence.
>
> 5. **There are numerous grammatical errors, typos, and formatting inconsistencies in this paper. The authors are advised to proofread the manuscript carefully. For example, the citation formats are inconsistent, some sentences lack ending punctuation, and there are obvious errors in the algorithm descriptions.**
>
> Response: We have thoroughly proofread the manuscript and corrected the grammatical errors, typographical mistakes, and formatting inconsistencies that we identified. We also reviewed and revised the algorithm description to improve its accuracy and clarity.
>
> Thank you again for your valuable comments and suggestions. We sincerely appreciate the time and effort you devoted to reviewing our manuscript. We hope that the revisions have addressed your concerns. If any part of the revised manuscript remains unclear, we would be grateful for your further feedback.

---

> > ### Comment · Reviewer_1sVS · 2026-07-08
> >
> > I would like to thank the authors for their efforts and detailed responses. While the revised manuscript has indeed seen significant improvements, I believe it still falls short of the criteria for acceptance. I recommend that the authors further refine the presentation of the manuscript. For instance, they should consider removing some unnecessary content, assigning more formal names to modules such as 'high-level constraint reward' and 'high-level policy,' and providing high-quality vector graphics for the experimental results.

---

> > > ### Author Response · Authors · 2026-07-08
> > >
> > > Thank you for these additional recommendations.
> > >
> > > We have updated the figures presenting the experimental results as vector graphics, improving their resolution when zoomed in.
> > >
> > > Regarding the terminology, we carefully considered your suggestion to adopt more formal names for the modules. However, we decided to retain the term "high-level policy" because it is the standard term used in the GCHRL literature for the policy learned by the high-level agent. Likewise, we retained "high-level constraint reward" because it explicitly describes the function of this auxiliary reward. We believe that alternative names that may appear more formal, such as high-level regulations, are less descriptive and could obscure both the module's purpose and the motivation behind its design.
> > >
> > > Regarding the unnecessary content, we recognize that opinions may differ on what is essential to the main contribution of the paper. We would greatly appreciate it if you could indicate which sections or specific parts you consider unnecessary. Your feedback would help us further optimize the manuscript's structure and improve its overall presentation.

---

### Review · Reviewer_1GPz · 2026-06-26

**Summary Of Contributions:**

The paper proposes Graph-Guided Quasimetric Dense Reward (G2QDR), a plug-in method for goal-conditioned hierarchical reinforcement learning. The method constructs a directed state graph online, trains a state connectivity model on ordered state pairs, and converts predicted connectivity into auxiliary dense rewards for high-level subgoal selection and low-level goal reaching. It also introduces an asymmetric penalty for transitions that are easy in one direction but hard to reverse, and uses a scheduled coefficient to anneal the dense reward because the shaping is not potential-based.

The method is evaluated on sparse-reward MuJoCo-style tasks, including AntMaze, AntGather, AntPush, AntFall, and Pusher, with four GCHRL backbones: HIRO, HRAC, HESS, and HLPS. The paper compares against the original backbones and an undirected graph baseline, G4RL, and includes ablations over the reward components, computational cost, and dense-reward schedule.

**Audience:**

Yes

**Audience Explanation:**

Yes. The paper would likely interest researchers working on hierarchical RL, goal-conditioned RL, graph-based exploration, sparse-reward control, and reward shaping. The core idea of using directed online state connectivity as a dense signal is worth considering. My concern is not that the work is uninteresting, but that the current manuscript needs a clearer fixed-protocol evaluation and a more direct validation of the connectivity model.

**Broader Impact Concerns:**

I do not see broader-impact concerns requiring rejection. The work is methodological and evaluated in simulated control environments. If the authors discuss real-world robotic deployment, a short note on safety risks from miscalibrated learned rewards or irreversible transitions would be useful.

**Claims And Evidence:**

No

**Claims Explanation:**

The idea is promising, but the current evidence does not yet fully support the paper's main claims.

My main concern is the relationship between Table 1 and Table 2. Table 1 reports one `+G2QDR` result for each backbone and environment, while Table 2 defines four variants: high-level reward only, low-level reward only, combined rewards, and full G2QDR with the asymmetric penalty. The Table 1 `+G2QDR` numbers appear to match the best variant from Table 2 separately for each cell, rather than one fixed G2QDR configuration. I verified that all 24 `+G2QDR` cells in Table 1 equal the maximum over the four ablation variants, and 15 of the 24 do not match the full variant with penalty. If Table 1 is using this per-cell selection, the main comparison is not a fair evaluation of a fixed algorithm. If not, the authors should explain why the numbers align in this way.

A second issue is that the central mechanism is not directly validated. The paper interprets the learned score `C_theta(psi(s_u,s_v))` as connectivity, transition ease, or directional accessibility, but it is trained on adjacency weights derived from recent trajectory windows. This target may mix transition feasibility with visitation frequency under the current policy. I would like to see a direct diagnostic showing that the learned score, or the asymmetry gap `C(s_u,s_v)-C(s_v,s_u)`, correlates with empirical reachability, first-passage time, directional irreversibility, or held-out graph connectivity. Without such evidence, the directed-connectivity claim is plausible but not yet clearly established.

Third, several claims should be narrowed or better supported. The method is called "quasimetric," but the model is an unconstrained directed similarity score and does not enforce or evaluate quasimetric properties such as triangle inequality or non-negativity. The sample-efficiency claim is also not fully demonstrated, since the main table reports final performance rather than learning curves or area under the curve. Similarly, the "minimal computational overhead" claim should be softened: the reported HLPS+G2QDR settings appear to cost about 1.4x to 1.9x the vanilla runtime, which may be acceptable but is not obviously minimal.

Finally, some implementation details needed for reproduction are missing or ambiguous. The state representation `phi`, the schedule boundaries for `lambda`, the exact training budget, and the meaning of the reported uncertainty should be specified. Algorithm 1 also appears to contain a small but important typo, storing `[r_h, r_l, r_hp, r_hp]` instead of including `r_lp`.

**Requested Changes:**

1. Clarify and, if necessary, redo the main comparison. Define exactly what `+G2QDR` means in Table 1. If it is a per-environment best choice among ablation variants, please report a fixed, pre-specified G2QDR configuration as the main method, with fair tuning for baselines.

2. Add a direct connectivity/asymmetry diagnostic. For example, evaluate whether `C_theta` predicts held-out reachability or whether `C(s_u,s_v)-C(s_v,s_u)` tracks known asymmetric structure in AntPush, AntFall, or Pusher. Alternatively, soften the causal claim about directed quasimetric structure.

3. Align claims with evidence. Please either add learning curves/statistical tests/runtime breakdowns or avoid strong language such as "significantly enhances," "sample efficiency," and "minimal overhead." Also consider reframing "quasimetric" as "directed/asymmetric connectivity" unless quasimetric structure is explicitly modeled or evaluated.

4. Improve reproducibility. Specify `phi`, the `lambda` schedule values, the training/evaluation protocol, what `+/-` denotes, and how high-level rewards are aggregated over low-level steps. Fix the Algorithm 1 reward tuple.

---

> ### Author Response · Authors · 2026-07-02
> **Reply to reviewer 1GPz**
>
> Thank you for the time and effort you dedicated to reviewing our manuscript, as well as for your insightful comments and suggestions. We have carefully revised the manuscript in response to your recommendations. Below, we provide a point-by-point response outlining the changes we have made.
>
> 1. **Clarify and, if necessary, redo the main comparison. Define exactly what +G2QDR means in Table 1. If it is a per-environment best choice among ablation variants, please report a fixed, pre-specified G2QDR configuration as the main method, with fair tuning for baselines.**
>
> Response: We were indeed performing a per-environment selection of the best-performing variant in the original manuscript. In the revised version, we have clarified the meaning of “+G2QDR” in the caption of Table 1. In addition, we now report the different G2QDR variants as separate entries in the table to ensure a clearer and more transparent comparison.
>
> 2. **Add a direct connectivity/asymmetry diagnostic. For example, evaluate whether C_theta predicts held-out reachability or whether C(s_u,s_v)-C(s_v,s_u) tracks known asymmetric structure in AntPush, AntFall, or Pusher. Alternatively, soften the causal claim about directed quasimetric structure.**
>
> Response: We have added an asymmetry measure in Appendix D.1 and, based on this, conducted a direct evaluation of how the asymmetry gap \(C(s_u, s_v) - C(s_v, s_u)\) evolves across different environments in Appendix D.2. Please refer to the relevant sections for further details.
>
> 3. **Align claims with evidence. Please either add learning curves/statistical tests/runtime breakdowns or avoid strong language such as "significantly enhances," "sample efficiency," and "minimal overhead." Also consider reframing "quasimetric" as "directed/asymmetric connectivity" unless quasimetric structure is explicitly modeled or evaluated.**
>
> Response: We have revised the wording of our claims in the abstract, introduction, and experimental sections to better align with the empirical results. For example, we have softened several statements, replacing “significantly enhances” with “generally enhances,” “sample efficiency” with “learning efficiency,” and “minimal overhead” with “acceptable overhead.” We have also reframed “quasimetric” as “transition asymmetry” throughout the manuscript.
>
> 4. **Improve reproducibility. Specify phi, the lambda schedule values, the training/evaluation protocol, what +/- denotes, and how high-level rewards are aggregated over low-level steps. Fix the Algorithm 1 reward tuple.**
>
> We have addressed the reproducibility concerns by adding or clarifying several key implementation details. The state representation function ϕ is now explicitly defined in Appendix C as a feature selection function that maps raw states to a reduced representation by retaining only location-related features (e.g., positions of the agent, objects, and goals) while discarding other variables such as angular velocities, joint angles, and other kinematic information.
>
> We have also clarified the training and evaluation protocol in Section 4.2. Specifically, agents are trained for 20,000 episodes in each environment, with evaluation performed every 1,000 training episodes over 100 independent trials without further learning. Performance is reported as success rate, defined as the percentage of successful task completions, and all results are averaged over 10 independent runs and reported as mean $\pm$ standard deviation.
>
> In addition, we have clarified the reward structure in Appendix B.1. For these environments, no reward information is exchanged between the high-level and low-level agents; therefore, no aggregation of low-level rewards is performed. As the environment provides only a terminal sparse reward, each high-level transition receives the auxiliary reward generated by our framework, in addition to the terminal environment reward.
>
> Finally, Algorithm 1 has been carefully reviewed and rewritten to correct and improve the reward specification and overall clarity.
>
> We hope our responses have addressed your concerns. Please do not hesitate to let us know if any point requires further clarification.

---

> > ### Comment · Reviewer_1GPz · 2026-07-10
> > **Response to the authors: revision verified, main concerns resolved, three small items remain**
> >
> > Thank you for the detailed point-by-point response and for the revision. I have checked the revised manuscript. The main concerns from my review are resolved. I go through my four requested changes in order.
> >
> > 1. Main comparison protocol
> >
> > I appreciate the authors' direct acknowledgment that the original Table 1 reported a per-environment selection of the best-performing variant. The revision goes further than the response letter describes: the oracle-selected `+G2QDR` row has been removed entirely and replaced by two fixed, pre-specified configurations, `+G2QDR w/o PT` and `+G2QDR w/ PT`. I verified that these rows match the variant-(c) and variant-(d) entries of Table 2 exactly, for all four backbones, and that the conclusions in Sec. 4.2 have been rewritten accordingly.
> >
> > 2. Direct connectivity
> >
> > Fig. 5) constitute the direct evaluation I requested, and the growing separation supports the claim that the model captures directional structure rather than only improving downstream returns.
> >
> > However, Table 5 introduces an internal inconsistency that should be fixed: AntGather receives an asymmetry score of 0.42 — higher than Pusher (0.39) and close to AntFall (0.48) — yet Sec. 4.1 still characterizes AntMaze and AntGather as "largely symmetric," and D.1 states that the scores "align with the qualitative discussion." These statements do not fit together, and the symmetric/asymmetric grouping underlies the "gains are more pronounced in asymmetric tasks" narrative.
> >
> > 3. Claims aligned with evidence
> > I confirm the softened wording in the revision: "generally enhances," "learning efficiency," "acceptable computational overhead," and the revised contribution bullet separating theoretical compatibility from the four evaluated backbones. Two residuals: (a) the abstract still states that G2QDR "can be integrated into any existing GCHRL architecture to boost performance," which is stronger than what is demonstrated — please align this sentence with the revised contribution wording; (b) the method name retains "Quasimetric" (abstract and Sec. 3) although the model, as the authors now state, predicts empirical connectivity and enforces no quasimetric properties. Keeping "quasimetric environments" for the setting (with the citations) is fine; for the method name, I suggest either renaming or a one-line disclaimer that the name refers to the motivating setting rather than to a quasimetric model. This last point is a suggestion, not a condition.
> >
> > 4. Reproducibility
> >
> > Verified in the revision: $\phi$ is now defined (Appendix C: feature selection retaining location-related features); the training/evaluation protocol is specified (20,000 episodes; evaluation every 1,000 episodes over 100 trials; mean $\pm$ standard deviation over 10 runs); high-level reward handling is clarified (Appendix B.1: no aggregation of low-level rewards; terminal sparse reward plus the auxiliary term); and Algorithm 1 has been rewritten correctly — the $\lambda \neq 0$ branch now stores both $r_{hp}$ and $r_{lp}$, the $\lambda = 0$ branch is consistent, and the schedule symbol is unified to $\lambda$ across Sec. 3.4 and the algorithm. The new hyperparameter sensitivity study (Appendix D.3, over $W$, $p$, $\epsilon_d$, and the reward weights on two backbones) is a welcome addition.
> >
> > The one explicitly requested item that is still missing: the numerical values of the $\lambda$-schedule boundaries $n_1$–$n_4$ appear nowhere in the revision. Every reported experiment — and the schedule ablation in Sec. 4.4 / Fig. 4 — depends on them. Please add the values used in the experiments to the hyperparameter appendix (per environment/backbone if they vary), with a sentence on how they were chosen.
> >
> > One new clarification: the revised Table 4 lists the novelty threshold as $\epsilon_d = 0.5$ for AntMaze and $1$ for the other environments, whereas the original submission listed $0.1$/$0.2$, while the Table 1–2 results are unchanged between versions. Finally, two cosmetic nits: Algorithm 1 stores $-\|\phi(s_{t+1}) - g_t\|^2$ while Eq. (9) uses the unsquared $-\|\phi(s_{t+1}) - g_t\|_2$ (and Sec. 2.2 the squared $\|\cdot\|_2^2$) — please make the norm consistent.

---

> ### Author Response · Authors · 2026-07-10
>
> Thank you for your additional comments.
>
> In response to your new concerns, we have made the following revisions to the manuscript:
>
> 1. **However, Table 5 introduces an internal inconsistency that should be fixed**
>
> We have revised the sentences in Section 4.1 as follows: "**AntMaze** is largely symmetric, as the agent can relatively easily return to previously visited states. In contrast, **AntGather**, **AntPush**, **AntFall**, and **Pusher** exhibit different types of asymmetry: reversing object displacement in AntGather, AntPush and Pusher is considerably more difficult, and ascending cliffs in AntFall is substantially harder than descending them. " We also changed our statement in D.1 as: "The estimated asymmetry scores are consistent with the qualitative discussion in Section 4.2: the directed G2QDR is more likely to outperform its undirected counterpart, G4RL, on **AntGather**, **AntPush**, **AntFall**, and **Pusher**, supporting our characterization of these environments as exhibiting greater structural asymmetry."
>
> 2. **Two residuals regarding wording**
>
> We have modified the abstract as: "We demonstrate that our proposed framework, Graph-Guided Quasimetric Dense Reward (G2QDR), can theoretically be integrated into any existing GCHRL architecture, and the state connectivity model is efficiently implemented via a neural network trained on a directed state graph generated during exploration." to align with the revised contribution wording.
>
> We also added "The term 'quasimetric' in the method name does not imply that our approach explicitly or implicitly employs a quasimetric model. Rather, it emphasizes the underlying motivation of the method: enabling the agent to distinguish between opposite directions of transition between states." as a one-line disclaimer in a footnote on page 4.
>
> 3. **the numerical values of the $\lambda$ missing**
>
> We added the numerical values of $n_1$ to $n_4$ in Table 4 with the explanation "The hyperparameters $n_1$ through $n_4$ are chosen such that $n_1$ and $n_2$ fall within the early stage of training, while $n_3$ and $n_4$ correspond to the late stage. All experiments are conducted using this single choice of $n_1$ to $n_4$, and alternative configurations were not explored." in the text below the table.
>
> 4. **Regarding the new clarification**
>
> We confirmed that the main results in Tables 1 and 2 were obtained using $\epsilon_d = 0.5$ for AntMaze and $\epsilon_d = 1$ for the other environments. The values $0.1/0.2$ reported in the original version were residuals from an earlier pre-submission draft. We identified this issue while conducting the hyperparameter ablation studies. We apologize for overlooking this in the original submission and for any confusion it may have caused.
>
> For the norm, we now consistently use the unsquared notation $ -|\phi(s_{t+1})-g_t|_2 $ (the Euclidean norm) throughout the manuscript, which is consistent with our experimental settings and previous literature.
>
> Thank you again for pointing these out. Please let us know if our responses adequately address your questions.

---

### Review · Reviewer_tkU5 · 2026-06-26

**Summary Of Contributions:**

This paper introduces G2QDR, a graph-guided dense reward framework for goal-conditioned hierarchical reinforcement learning. The method incrementally constructs a directed state graph during exploration, trains an order-sensitive neural connectivity model to estimate directed pairwise state connectivity, and converts these connectivity estimates into auxiliary rewards for both high-level subgoal selection and low-level goal-reaching. The motivation is that undirected graph-based GCHRL methods may fail to capture quasimetric or asymmetric transition structure.

The main strengths are the clear motivation for modeling directed connectivity, the integration of the proposed module into multiple GCHRL backbones, and a reasonably broad experimental section including comparisons to an undirected graph variant, reward-component ablations, runtime/performance trade-offs, and dense-signal schedule ablations. The results suggest that G2QDR often improves over vanilla GCHRL baselines and can be particularly helpful in tasks with asymmetric transition dynamics.

The main weaknesses are that the key connectivity model is evaluated only indirectly through downstream RL performance, without direct evidence that it accurately captures directed reachability or transition feasibility. The method also relies on several heuristic graph-construction and reward-design choices, introduces many hyperparameters, and the asymmetric penalty has task-dependent behavior. Some implementation details are underspecified, especially regarding replay-buffer rewards, high-level transition construction, and the non-stationarity of the learned dense reward. The reported computational overhead is also non-negligible, suggesting that claims about minimal overhead should be qualified.

**Audience:**

Yes

**Audience Explanation:**

Yes. I believe at least some individuals in the TMLR audience would be interested in the findings of this paper, especially readers working on goal-conditioned reinforcement learning, hierarchical RL, graph-based RL, sparse-reward learning, and reward shaping.

The paper addresses a relevant problem: how to exploit structured state connectivity information in goal-conditioned hierarchical reinforcement learning, particularly when the environment dynamics are not well captured by symmetric or undirected notions of distance. The idea of using directed connectivity estimates to shape both high-level subgoal selection and low-level goal-reaching behavior is potentially useful for researchers interested in long-horizon sparse-reward tasks and learned topological abstractions of state spaces.

I also think the empirical findings are of interest beyond the specific proposed method. The comparison between directed and undirected graph-based variants is relevant to the broader question of when asymmetric reachability matters in RL. The ablations over high-level rewards, low-level correction terms, and asymmetric penalties are informative: they suggest that connectivity-based dense rewards can be beneficial, but also that the asymmetric penalty is task-dependent and may not be universally useful. Similarly, the dense-signal scheduling experiments provide a practical lesson that auxiliary rewards derived from an evolving learned model may need to be delayed and annealed, rather than applied throughout training.

**Broader Impact Concerns:**

No major broader impact concerns.

**Claims And Evidence:**

No

**Claims Explanation:**

I would answer this question only partially supported. The experimental results are promising and support the claim that adding G2QDR can improve the final performance of several GCHRL baselines on the evaluated MuJoCo tasks. However, I do not think the evidence is yet sufficiently accurate, convincing, and clear for several of the paper’s central claims.

The main evidence supporting the paper is the comparative evaluation across four GCHRL backbones, where G2QDR often improves over the corresponding vanilla baselines. The comparison against the undirected graph variant also provides some evidence that directed connectivity can be useful, especially in the tasks that the authors characterize as more asymmetric. The ablation studies further suggest that the high-level connectivity reward and low-level correction reward both contribute to performance, and the schedule ablation supports the need for delayed activation and late-stage annealing of the dense signal.

However, the key mechanistic claim of the paper is not directly validated. The submission repeatedly motivates the method through quasimetric or asymmetric state-transition structure, but the learned connectivity model is evaluated only indirectly through downstream RL performance. There is no direct diagnostic showing that connectivity model accurately captures directed reachability, transition feasibility, shortest-path distance, forward/reverse success probability, or any ground-truth asymmetric structure. The environments are described qualitatively as having varying degrees of asymmetry, but this asymmetry is not quantified. As a result, it remains unclear whether the directed connectivity model is genuinely learning quasimetric transition structure, or whether the observed gains are mainly due to additional auxiliary rewards and regularization.

The definition of the connectivity target is also somewhat ambiguous. The model is trained to regress normalized graph edge weights, which are induced by online trajectories, temporal windows, and heuristic edge decay. This quantity appears to represent empirical temporal connectivity or visitation-derived reachability statistics, rather than a clearly defined transition probability, shortest-path distance, transition cost, or success likelihood. Since these graph statistics depend strongly on the current exploration policy, low connectivity may reflect insufficient exploration rather than true transition infeasibility.

The paper partially addresses the concern that dense rewards may bias the original task by introducing a stage-dependent $\lambda$ schedule and annealing the induced rewards to zero. This is a sensible practical design, and Figure 3 supports its usefulness. However, because the rewards are not potential-based, annealing does not provide a theoretical guarantee that the original optimal policy is preserved; the shaped rewards still affect the data distribution, replay buffer, and policy trajectory during training.

Some claims should therefore be weakened. In particular, the claim of minimal computational overhead is not well supported by the reported runtime ratios of roughly $1.4\times$ to $1.9\times$ relative to vanilla HLPS. I would also suggest avoiding language implying that the model predicts “true” transition feasibility unless this is directly evaluated.

**Requested Changes:**

1. The paper should provide direct diagnostics showing whether $C_\theta(s_u,s_v)$ actually captures directed reachability, for example by comparing it with empirical success probabilities, forward/reverse reachability, graph shortest-path distance, or ground-truth reachability in a controlled asymmetric environment. This is important because the central mechanism is currently validated mostly through downstream RL performance.

2. The paper should quantify the asymmetry of the evaluated environments. G2QDR is motivated by quasimetric or asymmetric transitions, but the asymmetry of AntPush, AntFall, and Pusher is mostly described qualitatively. Reporting explicit measures such as forward/reverse reachability gaps or distributions of $C(s_i,s_j)-C(s_j,s_i)$ would make the motivation and empirical claims more convincing.

3. The authors should clarify what $C_\theta$ is actually trained to predict. Since the supervision target is a normalized trajectory-induced edge weight, the model appears to estimate empirical temporal connectivity rather than a clearly defined transition probability, reachability probability, shortest-path distance, transition cost, or success likelihood. The paper should define this quantity precisely and avoid implying that it predicts true transition dynamics unless directly supported.

4. The authors should specify whether dense rewards are stored at data-collection time or recomputed during off-policy updates as $C_\theta$ and $\lambda$ change. They should also clarify how high-level transitions are constructed when subgoals are selected every $K$ primitive steps, and whether high-level rewards are accumulated over the full $K$-step interval.

5. In Algorithm 1, the $\lambda=0$ branch appears to store a value of 1 in the position where the other branch stores $\lambda$. This seems inconsistent and should be corrected or clearly explained.

6. Several claims should be softened to better match the evidence. In particular, "compatible with any GCHRL algorithm" should be revised to reflect that the method is evaluated on four backbones; "minimal computational overhead" should be weakened because the reported runtime overhead is non-trivial; and "consistent improvements" should be qualified because gains over the undirected variant and the asymmetric penalty are task-dependent.

7. The paper should better discuss the limitation that the induced dense rewards are not potential-based. Although the proposed annealing schedule is a reasonable empirical mitigation, it does not guarantee preservation of the original optimal policy, since the shaped rewards still affect the data distribution, replay buffer, and training trajectory.

8. The paper should include sensitivity analysis for key heuristic choices and hyperparameters. The method depends on graph size $N$, novelty threshold $\epsilon_d$, temporal window $W$, edge-decay exponent $p$, reward weights, and the $\lambda$ schedule. At minimum, sensitivity to the most influential choices, especially $W$, $p$, $\epsilon_d$, and reward weights, should be reported.

9. The limitations section should be expanded. It currently focuses mainly on hyperparameter tuning and computational cost, but it should also discuss exploration bias versus true infeasibility, the possibility that learned connectivity reinforces early exploration biases, the task dependence of the asymmetric penalty, heuristic graph construction, dependence on the state representation, and the lack of direct validation of directed reachability.

---

> ### Author Response · Authors · 2026-07-02
> **Reply to reviewer tkU5 (1/2)**
>
> Thank you for your time devoted to our work and for your valuable suggestions. Below is a point-by-point summary of our revisions addressing your comments.
>
> 1. **The paper should provide direct diagnostics showing whether $\mathcal{C}_\theta$ actually captures directed reachability, for example by comparing it with empirical success probabilities, forward/reverse reachability, graph shortest-path distance, or ground-truth reachability in a controlled asymmetric environment. This is important because the central mechanism is currently validated mostly through downstream RL performance.**
>
> Response: We have added a direct diagnostic analysis of how the asymmetry gap, $ \mathcal{C}(s_u, s_v) - \mathcal{C}(s_v, s_u) $, evolves across different environments in Appendix D.2. We also describe the metrics used and provide justification for why this serves as a meaningful diagnostic. Please refer to the revised manuscript for further details.
>
> 2. **The paper should quantify the asymmetry of the evaluated environments. G2QDR is motivated by quasimetric or asymmetric transitions, but the asymmetry of AntPush, AntFall, and Pusher is mostly described qualitatively. Reporting explicit measures such as forward/reverse reachability gaps or distributions of $ \mathcal{C}(s_u, s_v) - \mathcal{C}(s_v, s_u) $ would make the motivation and empirical claims more convincing.**
>
> Response: Following the previous response, we define an asymmetric metric based on rule-based criteria in Appendix D.1. Please refer to that section for a detailed description.
>
> 3. **The authors should clarify what $\mathcal{C}_\theta$  is actually trained to predict. Since the supervision target is a normalized trajectory-induced edge weight, the model appears to estimate empirical temporal connectivity rather than a clearly defined transition probability, reachability probability, shortest-path distance, transition cost, or success likelihood. The paper should define this quantity precisely and avoid implying that it predicts true transition dynamics unless directly supported.**
>
> Response: We have clarified in the introduction that our method predicts "empirical connectivity between states based on observed arrivals". We have also revised the wording throughout the paper to avoid implying that the model is trying to estimate "true transition dynamics".
>
> 4. **The authors should specify whether dense rewards are stored at data-collection time or recomputed during off-policy updates. They should also clarify how high-level transitions are constructed when subgoals are selected every $K$ primitive steps, and whether high-level rewards are accumulated over the full $K$-step interval.**
>
> Response: Dense rewards are stored at data-collection time. We further clarify in Appendix B.1 that no reward information is exchanged between the high-level and low-level agents, and therefore no aggregation of low-level rewards is performed. Since the environment provides only a terminal sparse reward, each high-level transition receives the auxiliary reward generated by our framework in addition to the terminal environment reward.
>
> 5. **In Algorithm 1, the $\lambda=0$ branch appears to store a value of 1 in the position where the other branch stores $\lambda$. This seems inconsistent and should be corrected or clearly explained.**
>
> Response: Yes, this was a typographical error. We have carefully reviewed and rewritten Algorithm 1 to correct this issue and improve both the reward specification and overall clarity.
>
> 6. **Several claims should be softened to better match the evidence. In particular, "compatible with any GCHRL algorithm" should be revised to reflect that the method is evaluated on four backbones; "minimal computational overhead" should be weakened because the reported runtime overhead is non-trivial; and "consistent improvements" should be qualified because gains over the undirected variant and the asymmetric penalty are task-dependent.**
>
> Response: We have revised the wording of our claims in the abstract, introduction, and experimental sections to better align with the empirical results. The statement “compatible with any GCHRL algorithm” has been revised to “Our architecture is theoretically compatible with any GCHRL algorithm in both symmetric and asymmetric environments. In our experiments, we evaluate it with four representative backbone methods (HIRO, HRAC, HESS, and HLPS), and observe performance improvements across a broad range of tasks.” We have also changed “minimal computational overhead” to “acceptable computational overhead” and avoided overly strong claims such as “consistent” or “significant” where not fully supported by the results.

---

> ### Author Response · Authors · 2026-07-02
> **Reply to reviewer tkU5 (2/2)**
>
> 7. **The paper should better discuss the limitation that the induced dense rewards are not potential-based. Although the proposed annealing schedule is a reasonable empirical mitigation, it does not guarantee preservation of the original optimal policy, since the shaped rewards still affect the data distribution, replay buffer, and training trajectory.**
>
> Response: We have added a discussion of this limitation in the Limitations section, noting that the induced dense rewards are not potential-based and therefore may not preserve the original optimal policy.
>
> 8. **The paper should include sensitivity analysis for key heuristic choices and hyperparameters. **
>
> Response: We have conducted ablation studies on several key hyperparameters, along with empirical analysis and interpretation of the results. Please refer to Appendix D.3 for further details.
>
> 9. **The limitations section should be expanded. It currently focuses mainly on hyperparameter tuning and computational cost, but it should also discuss exploration bias versus true infeasibility, the possibility that learned connectivity reinforces early exploration biases, the task dependence of the asymmetric penalty, heuristic graph construction, dependence on the state representation, and the lack of direct validation of directed reachability.**
>
> Response: We have expanded the limitations sections as follows: Despite the promising advantages demonstrated by G2QDR in hierarchical reinforcement learning (HRL) tasks, several limitations remain. First, its performance is highly sensitive to a number of key hyperparameters, including $\epsilon_d$, the significance-related hyperparameters ($\alpha$), $p$, and $W$. Obtaining strong and stable performance across diverse environments therefore requires careful and often time-consuming manual tuning, which may limit the scalability and practical applicability of the method.
>
> Second, the graph is constructed dynamically during exploration and is consequently influenced by the behavior policy of the underlying backbone method. Such policy-dependent graph construction introduces exploration bias, potentially resulting in an incomplete or unrepresentative approximation of the environment structure. As a consequence, the dense rewards induced from the learned graph may inherit this bias, which could affect the quality of guidance provided to the agent.
>
> Third, the environments considered in our experiments are predominantly continuous and characterized by high-dimensional state representations. In these settings, the ground-truth state connectivity structure and the true transition distances between states are generally unavailable. Therefore, directly evaluating the quality of the constructed graph against an optimal or reference graph is infeasible. Instead, the effectiveness of the learned graph is evaluated indirectly through its impact on downstream task performance.
>
> Finally, the introduced auxiliary reward is not potential-based, indicating that it alters the optimization objective and changes the optimal policy. Although we introduce a dense signal scheduling strategy to mitigate this issue, the auxiliary reward may still influence exploration bias and the replay buffer, which could potentially affect the performance of our method.
>
> Thank you again for your constructive comments. If we have overlooked anything or if there are still any ambiguities in the revised manuscript, please let us know.

---

> > ### Comment · Reviewer_tkU5 · 2026-07-08
> >
> > Thank you for the revisions. I appreciate the added asymmetry analysis and the clarification of several implementation details. However, I still have two remaining concerns.
> >
> > First, the new diagnostic in Appendix D.2 evaluates the separation of the absolute asymmetry gap between rule-based symmetric and asymmetric pairs. While this is useful, it mainly shows that the learned scores exhibit directional differences; it does not directly establish that $C_\theta(s_i,s_j)$ accurately predicts directed reachability or transition feasibility. Could the authors clarify whether they can compare $C_\theta$ with an empirical reachability proxy, such as forward/reverse rollout success probabilities between sampled state pairs, even on a smaller controlled subset of states or environments?
> >
> > Second, I remain unclear about the high-level transition construction. The revision states that low-level rewards are not aggregated and that each high-level transition receives the auxiliary reward plus the terminal sparse reward. However, since the high-level policy selects a subgoal every $K$ primitive steps, please clarify whether high-level transitions are constructed over the full $K$-step interval, what the high-level next state is, and whether the high-level auxiliary/environment rewards are accumulated over this interval or assigned at each primitive step.

---

> ### Author Response · Authors · 2026-07-09
>
> We thank the reviewer for the additional comments. To further verify that $\mathcal{C}_\theta(\cdot)$ captures transition feasibility beyond merely exhibiting directional differences, we added a new diagnostic analysis in Appendix D.2 following the directional difference analysis. Specifically, we compare the ordering induced by the learned connectivity scores against an independent reachability proxy in AntMaze. Since AntMaze contains a single movable agent with nearly deterministic and reversible dynamics, the graph shortest-path distance between states provides a reliable approximation of empirical transition feasibility. We discretize the environment into a grid graph and evaluate whether the learned connectivity scores preserve the ordering of state-pair distances through a monotonicity preservation metric. The results show that the learned scores become increasingly aligned with the underlying transition feasibility structure during training. Details can be found on pages 20–21.
>
>
> The high-level transition is indeed constructed over the full $K$-step interval. Specifically, when the high-level policy receives the state $s_t$ at a decision point $t=1$, it outputs an action corresponding to a designated subgoal state that the low-level policy is expected to reach. The low-level policy then executes primitive actions for $K$ steps. After this execution period, the state actually reached by the agent, denoted as $s_{t+K}$, is used as the next state of the high-level transition.
>
> The high-level auxiliary reward is assigned once at the end of each $K$-step interval, rather than at every primitive step. The high-level environmental reward is obtained by accumulating the primitive environmental rewards over this same interval. Since the environmental reward is sparse, this accumulated reward is typically zero unless the agent reaches the task goal during the interval. Therefore, each high-level transition contains the auxiliary reward together with the accumulated sparse environmental reward over the corresponding $K$-step rollout.
>
> We hope this additional explanation addresses your concerns.